# A nanoscale reciprocating rotary mechanism with coordinated mobility control

Eva Bertosin [1,2], Christopher M. Maffeo[3,4], Thomas Drexler [1,2], Maximilian N. Honemann [1,2], Aleksei Aksimentiev [3,4] & Hendrik Dietz [1,2 ✉]

Biological molecular motors transform chemical energy into mechanical work by coupling cyclic catalytic reactions to large-scale structural transitions. Mechanical deformation can be surprisingly efficient in realizing such coupling, as demonstrated by the $F_1F_O$ ATP synthase. Here, we describe a synthetic molecular mechanism that transforms a rotary motion of an asymmetric camshaft into reciprocating large-scale transitions in a surrounding stator orchestrated by mechanical deformation. We design the mechanism using DNA origami, characterize its structure via cryo-electron microscopy, and examine its dynamic behavior using single-particle fluorescence microscopy and molecular dynamics simulations. While the camshaft can rotate inside the stator by diffusion, the stator's mechanics makes the camshaft pause at preferred orientations. By changing the stator's mechanical stiffness, we accelerate or suppress the Brownian rotation, demonstrating an allosteric coupling between the camshaft and the stator. Our mechanism provides a framework for manufacturing artificial nanomachines that function because of coordinated movements of their components.

[1] Lehrstuhl für Biomolekulare Nanotechnologie, Physik Department, Technische Universität München, Garching near Munich, Germany. [2] Munich Institute of Biomedical Engineering, Technische Universität München, Garching near Munich, Germany. [3] Department of Physics, University of Illinois at Urbana-Champaign, Urbana, IL 61801, USA. [4] Beckman Institute for Advanced Science and Technology, University of Illinois at Urbana-Champaign, Urbana, IL 61801, USA. ✉email: dietz@tum.de

Macroscopic machines commonly rely on a coordinated motion of multiple rigid components to perform their function. For example, an internal combustion engine uses a rotating camshaft to cyclically open or close the peripheral valves for fuel injection and exhaust gas removal; the coordination of the valves' operation is paramount to the engine's function. Nanoscale biological machines also often consist of multiple components that move in a coordinated fashion. For example, the rotation of the central shaft in $F_1F_O$ ATP synthase[1–3] produces cyclic structural transformations at the interfaces of the $F_1$ sub-units, coordinating cyclic chemical transformations. Intriguingly, the $F_1$ motor of $F_1F_O$ ATP synthase is reversible[4,5]: it can either function as a rotary motor powered by the chemical energy of ATP hydrolysis or it can use the mechanical energy of the central shaft rotation to catalyze synthesis of ATP. The fact that the $F_1$ ATP synthase can be both a motor and a chemical generator reflects the microscopic reversibility of elementary chemical processes and is a unique feature of molecular scale machines. Realizing a similar degree of mechanochemical coupling in a synthetic nanoscale system remains a landmark technological goal.

The construction of artificial molecular machines by chemical synthesis has previously provided important insights regarding how to create molecular mechanisms with internal degrees of freedom, such as catenanes and rotaxanes, and how to power molecular motions using chemical fuels, light, and other stimuli[6–12]. DNA nanotechnology has also already provided a range of mechanical systems including pivots, hinges, crank sliders, and rotary mechanisms[13–17] that can be reconfigured using strand displacement reactions (SDR)[18] or by changing environmental parameters such as pH, ionic strength, temperature, and external fields[19–24].

Whereas the molecular mechanisms generated by chemical synthesis tend to include on the order of 100 atoms, DNA nanostructures, in particular DNA origami objects, are much larger and can encompass hundreds of thousands of atoms[25–27]. As such, DNA origami nanomachines may offer additional opportunities for the assembly of mechanisms of coordinated mechanical power transmission. In this work, we describe the construction, computational characterization and experimental validation of a rotary mechanism with user-defined power and motion transmission. We conceive this object by combining macroscale machine design concepts with functional and structural aspects of the ATP synthase, and consider it to be a stepping stone toward creating artificial machines that achieve and generalize functionalities observed in biological motors.

## Results

**Design of the rotary mechanism**. We designed our rotary mechanism as a tetramer composed of a camshaft-like rotor in a surrounding stator. Rotations of the camshaft will induce reciprocal deformations of the structural elements in the stator (Fig. 1a and Supplementary Movie 1). We approximated the desired three-dimensional (3D) shapes of the components using the methods of multilayer DNA origami[28,29].

The stator comprises three similarly shaped components, consisting of 46 helices packed in parallel on a honeycomb-like lattice. One stator unit possesses an asymmetric feature (Supplementary Fig. 1) to discriminate the stator orientation relative to the shaft orientation by transmission electron microscopy (TEM) imaging. Each stator unit contains a rigid part (the "bearing") that will hold the shaft. The units also have two "pawls" that can flex in response to the camshaft rotation (Fig. 1b). The bearing and the pawls can be considered each as rigid blocks that are connected via two DNA double helices (the "support helices") that run vertically along the whole structure. The pawls are connected to the support helices via two crossovers at the top and bottom of the pawls. The

support helices can flex away from the central shaft, and the pawls can bend around the support helices to make room for the rotating camshaft. The pawls can also form base-pair stacking bonds at the blunt-ended helical interfaces between the bottom and top surface of pawls and bearing, respectively, which influences the flexibility of the pawls.

The camshaft consists of a shaft and a crossbar made of 24 helices packed in a honeycomb-like lattice (Fig. 1c and Supplementary Fig. 2). The cross-section of the shaft fits tightly into the central bore of the bearing (Supplementary Fig. 3). However, we also fixed a protruding feature on the shaft (the "cam") which clashes with the pawl helices, forcing them to flex away from the shaft. The cam and the crossbar mechanically trap the shaft inside the stator (Fig. 1d). This design may be considered as an analog of a rotaxane, with the stator being equivalent to the ring and the camshaft that of the dumbbell-axle.

For the assembly, we dock the camshaft first onto one stator unit before closing the full bearing (Fig. 1e) by hybridizing four staple strands protruding from the stator unit to a complementary single-stranded scaffold segment of the camshaft (Fig. 1e, top). The other two stator units can dock to each other via shape-complementary features carrying sticky ends (Fig. 1e, bottom). Once the complete DNA rotor complex is formed, we release the camshaft from its docking site via toehold-mediated strand displacement. To this end, invader strands are added that remove the initial strand linkages between the camshaft and the stator unit (Fig. 1f, top). Due to the mechanical interlocking, the camshaft remains sterically trapped in the stator (Fig. 1f, bottom), constrained to a rotary degree of freedom.

**Cryo-EM analysis of rotor structure and rotary motions**. We self-assembled the components of our rotary mechanism (Fig. 2a) using previously described methods[30] and determined suitable folding and assembly conditions using gel electrophoresis (Supplementary Figs. 4 and 5). We analyzed the structures of the resulting objects via negative stain TEM (Supplementary Fig. 6), and determined 3D cryo-electron microscopic (cryo-EM) density maps for each of the four DNA origami units of our complex (Fig. 2b–e and Supplementary Figs. 7–10), for the trimeric stator lacking the camshaft (Fig. 2f, g and Supplementary Fig. 11), and for the fully assembled tetrameric rotary mechanism including the camshaft (Fig. 3 and Supplementary Figs. 12–15).

**Structures of components**. The cryo-EM maps determined for the individual stator units agreed well with the design (Fig. 2b–e). The cryo-EM map determined for stator unit 1 had the highest resolution and featured regions where the grooves of the constituent DNA double helices can be discerned (Fig. 2b). We can distinguish several structural features in the 3D maps, such as the shape-complementary protrusions and recesses for docking the stator units into a complete bearing, the two support DNA double helices to which the pawls are anchored, and the asymmetric feature that marks stator unit 3 (Fig. 2d). In the cryo-EM map of the camshaft (Fig. 2e), we can recognize the honeycomb pattern, the protruding "cam" helices on the side of the shaft, and the crossbar. On the other end of the central shaft, we observe a small dent, which we assign to the scaffold segment that we left single-stranded for binding the camshaft to one of the stator units via staple strand hybridization.

**Empty stator vs stator with camshaft inside**. The 2D class averages (Fig. 2f) and the 3D cryo-EM map (Fig. 2g) that we determined for the empty stator reveal a structurally well-defined bearing that closely matches the design (Fig. 2g, slice 1, Supplementary Fig. 16). The asymmetric feature on stator unit 3 is

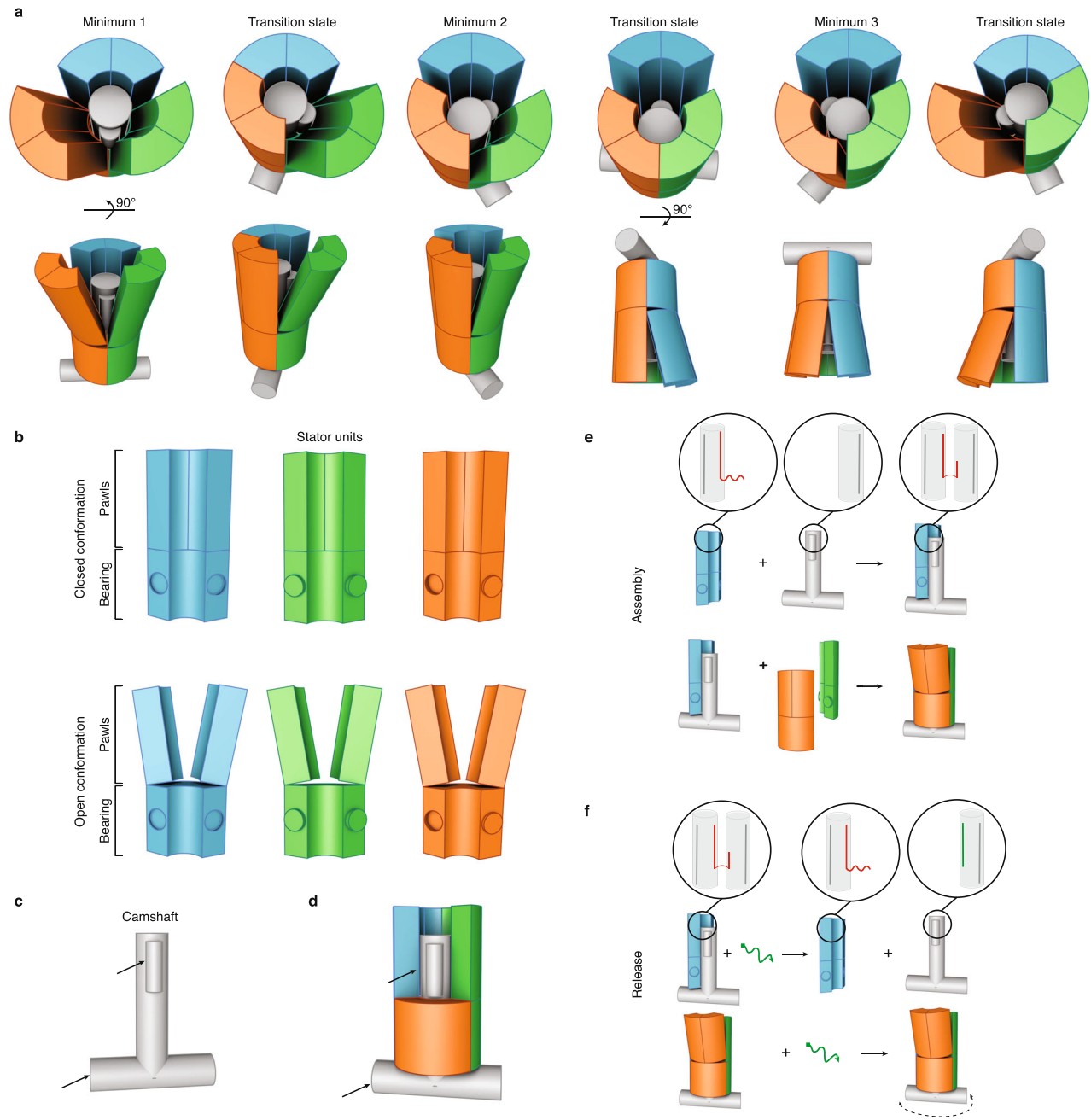

**Fig. 1 Conceptual design and assembly of the rotary mechanism. a** Sketches of the rotational mechanism in a top view (top) and side view (bottom). The shaft is depicted in gray, while the stator units are in blue, green, and orange. **b** Sketches of the stator units with pawls in the closed (top) and open (bottom) conformations. **c** Sketch of the camshaft. Black arrows indicate features used to prevent the camshaft from escaping the stator. **d** Sketch of the camshaft when trapped inside the stator. Pawls of the orange stator unit are not drawn. Black arrows as in **c**. **e** Sketch of the assembly steps for building the rotary mechanism. Red: connecting strands, gray: scaffold strand. **f** Top: sketch of the shaft release from the stator unit via toehold-mediated strand displacement. Bottom: same reaction but performed within the fully assembled stator. Green: invader strands.

clearly visible. By contrast, the signal from the pawls is more delocalized and fans away from the long axis of the stator (Fig. 2g, slice 2). Presumably, the loss of detail is due to conformational heterogeneity associated with flexing of the pawls. The top view (Fig. 2g, slice 3) shows that the pawls are rotated and displaced from their original position and that the central opening here is now smaller than near the bearing. When superimposing the map determined separately for the shaft on the map determined for the empty stator (Fig. 2h), we see that the shaft fits well inside the central bore of the stator in the bearing (slice 1) whereas the camshaft and the stator maps sterically clash in the region of

the pawls (slices 2 and 3). Therefore, the stator pawls need to be pushed outwards to accommodate the camshaft into the central bore. This is evident comparing the map of the empty stator with the map of the fully assembled complex (Supplementary Movie 2).

**Camshaft fixed in different orientations**. Our design was constructed such that placing the camshaft at different orientations into the stator should cause distinct shapes of the stator. To examine this feature, we prepared three distinct variants of the

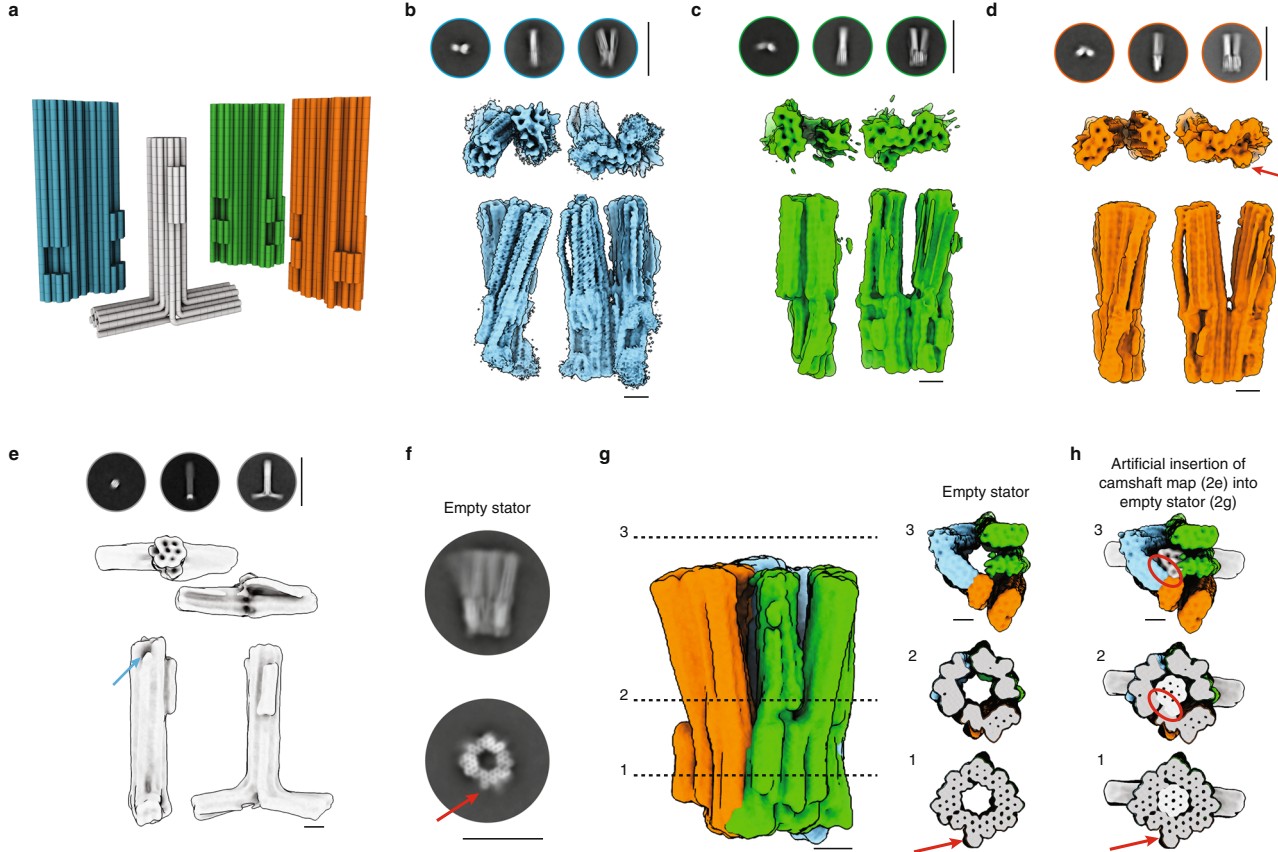

**Fig. 2 Cryo-EM analysis of the components. a** Cylinder models of our implementation with multilayer DNA origami. Cylinders represent DNA double helices. **b–e** Representative 2D class averages (top) and 3D electron density map (bottom), determined from cryo-EM micrographs of the three stator units and the camshaft. Red arrow: asymmetric feature on the stator unit 3. Blue arrow: single-stranded scaffold segment for successive binding to the stator units. See Supplementary Figs. 7–10. Scale bars of 2D classes: 100 nm. **f, g** Left to right: 2D class averages, 3D cryo-EM map, and cross-sectional slices of the 3D EM map of the stator assembled without camshaft. Red arrow indicates the asymmetric feature on stator unit 3. See Supplementary Fig. 11. Scale bar of 2D classes: 50 nm. **h** Slices as in **g**, but when the camshaft map from **e** is placed into the map of the empty stator from **g**. Red circles highlight clashes of the camshaft inside the empty stator. Red arrow indicates the asymmetric feature on stator unit 3. Scale bars of all 3D maps: 10 nm.

complete rotary mechanism with inserted shaft, in which the camshaft is initially fixed by staple strand linkages to stator units 1, 2, and 3, respectively. These variants thus realize three different, fixed orientations of the camshaft relative to the surrounding stator. We determined 3D cryo-EM maps for each of these variants (Fig. 3a). In the resulting maps we could discriminate the asymmetric feature present in the stator unit 3 and used it to assign the stator unit identities and to align the stator orientation. The camshaft indeed assumes three distinct positions inside the stator, rotated by 120°. These orientations can be discerned by comparing the orientation of the asymmetric feature in the stator (Fig. 3a, red arrows) relative to those of the camshaft crossbar (Fig. 3a, blue arrows). We note that the protruding cam on the camshaft can also be discerned in each of the three maps (Fig. 3a, yellow arrows). As designed, the cam is always oriented at 90° relative to the camshaft crossbar orientation. The cross-sectional slices reveal that at the level of the bearing (slice 1), the structures of the variants are all very similar. By contrast, the maps differ at the level of the pawls and seen from the top (slices 2, 3). The shape of the gaps between the six pawls and the central shafts depends on how the shaft is oriented relative to the stator, which was one of our design goals.

**Rotary motion**. To release the camshaft from the docking site we used toehold-mediated strand displacement (see "Methods" and

Supplementary Fig. 17). We acquired cryo-EM data of the rotary complex with the camshaft now free to rotate. 2D class averages already reveal crucial differences between the rotor complexes with a fixed camshaft (Fig. 3b) and the sample with presumably mobile camshaft (Fig. 3c). For instance, the honeycomb cross-section of the shaft, the cam, and the horizontal crossbar are clearly visible in the data with fixed shaft (Fig. 3b). On the other hand, these details are blurred in the sample with the released camshaft. The camshaft cross-section appears as a rotationally averaged version of a honeycomb (Fig. 3c). These images thus suggest that the camshaft is indeed rotating inside of the stator.

We further analyzed the cryo-EM data of the rotary complex with released camshaft using 3D classification. We found three dominant, structurally distinct 3D classes in the data set (Fig. 3d), containing 31k, 27k, and 20k particles, respectively. We aligned the stator of the maps using the asymmetric feature of stator unit 3 (Fig. 3d, red arrows). Each of the three 3D classes shows a different orientation of the T-crossbar of the shaft (Fig. 3d, blue arrows) relative to the asymmetric feature of the stator. The shaft-to-stator orientations are very similar to the samples with the camshaft bound to the stator (Fig. 3a). However, the cam of the camshaft could not be resolved in the 3D classes. The three 3D classes could thus each contain a mixture of particles featuring camshafts in two different orientations, rotated by 180°.

We also employed multibody refinement[31] to investigate the motion of the camshaft relative to the stator. To this end, the

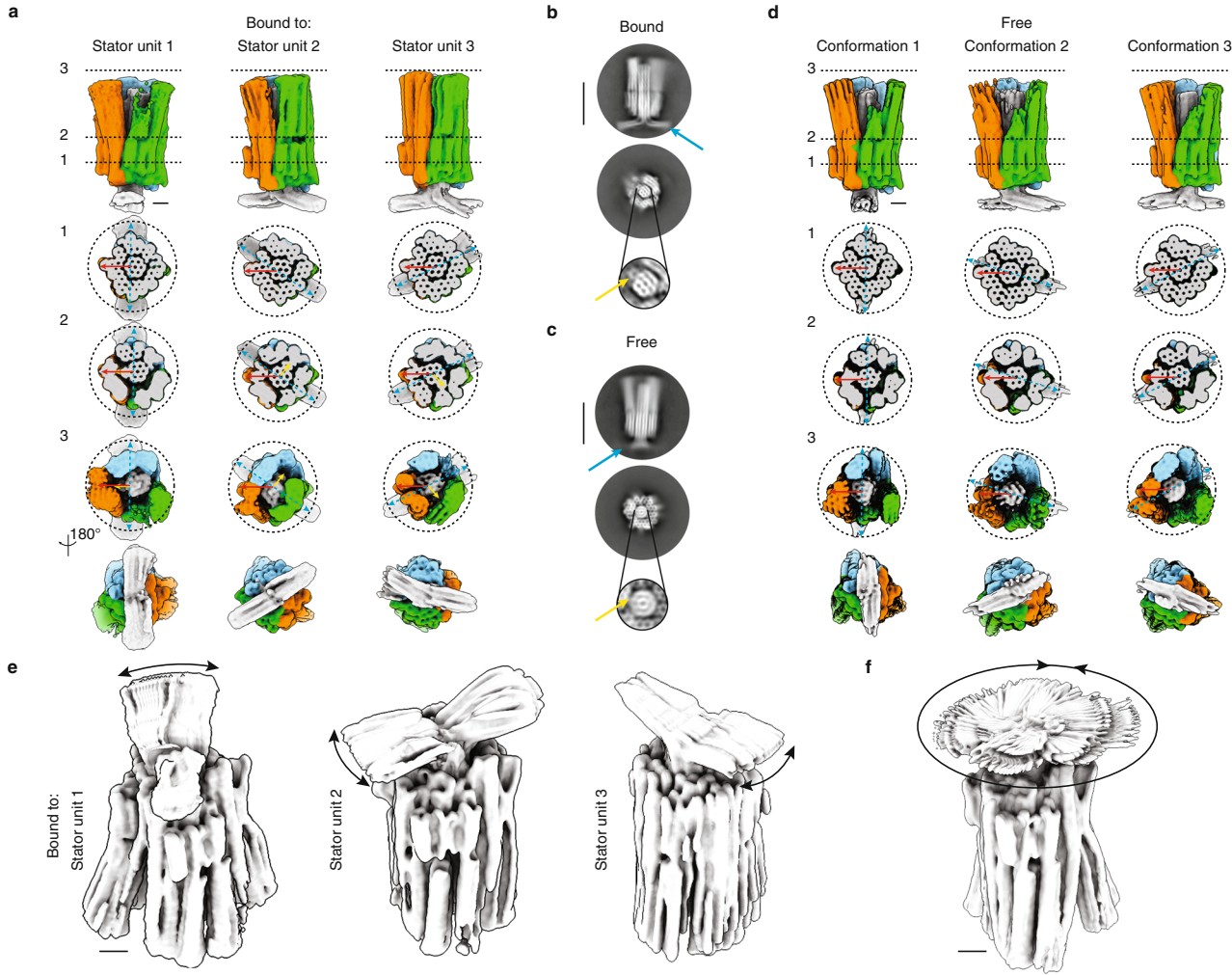

**Fig. 3 Cryo-EM analysis of the whole apparatus. a** Top: side view of the 3D cryo-EM maps determined separately for three variants of the mechanism, with the shaft docked to the stator unit 1, 2, or 3, respectively. See Supplementary Figs. 12–14. Bottom: cross-sectional slices through the maps. Red, blue, and yellow arrows highlight the asymmetric feature of the stator, the T-crossbar, and the cam, respectively. **b** 2D class averages of the mechanism with the shaft bound to stator unit 1. Blue and yellow arrows highlight crossbar and cam of the shaft, respectively. Scale bar: 50 nm. **c** 2D class averages of the mechanism determined from a sample where the shaft was released from the docking site. Blue and yellow arrows point to where crossbar and cam would be expected to be located, but the features are blurred. Scale bar: 50 nm. **d** Top: side views of three representative 3D classes discovered in the cryo-EM micrograph data determined from the same sample with the released camshaft. Bottom: cross-sectional slices through the 3D classes. Red arrows indicate the asymmetric feature of the stator, blue arrows show the position of the camshaft T-crossbar. The cam could not be resolved in the data. See Supplementary Fig. 15. **e** Overlay of frames from movies from multibody refinement analysis of data acquired separately, with the shaft docked to one stator unit 1, 2 or 3, respectively. Black arrows indicate the observed range of rotary motion of the shaft relative to the stator. **f** As in **e**, but for the sample where the shaft was released from its docking site. Scale bars of all 3D maps: 10 nm.

monomers are treated as rigid bodies that can move independently from each other. Using principal component analysis on the relative orientations of the bodies over all particle images in the data set, we computed movies for the important motions in the data. To illustrate these motions in a still image, we superimposed the frames of the resulting movies (Fig. 3e, f). For the three samples with fixed camshaft, we see that the dominant motion of the camshaft is restricted to some rotary wiggling with a ~20° range (Fig. 3e). By contrast, in the sample with the free camshaft, the rotations of the camshaft cover the entire 360° range (Fig. 3f). Together, the results from 3D classification and multibody analysis indicate that the camshaft can freely rotate inside the stator and has at least three, possibly six, preferred orientations. We also used the multibody analysis to study the flexibility of the pawls, treating each pawl as a separate rigid body. The displacement amplitudes of the pawls varied between ~2 and ~28 nm (Supplementary Fig. 18).

**Single-particle fluorescence measurements**. We used total internal reflection fluorescence microscopy (TIRFM) to study the dynamical behavior of our rotary complex. We anchored the stator to a glass slide covered by polyethylene glycol (PEG) and biotin by extending the stator unit 1 with a helix bundle (6hb) domain at the top of one of the pawls (Fig. 4a and Supplementary Figs. 19 and 20). Protruding from the 6hb are eight DNA adapter strands to which we hybridized biotinylated DNA strands (Supplementary Fig. 21) that anchor the stator in a multivalent attachment to the slide via biotin–neutravidin–biotin bridges. The multivalent binding is crucial to suppress rotations of the entire mechanism on the glass slide. The 6hb was labeled with 10 fluorescent dyes (cy5) to detect the position of the stator. We also extended the T-crossbar of the camshaft with a 10-helix bundle lever arm featuring 10 fluorescent dyes (cy3) at its tip, resulting in a ~290 nm long "pointer" (Fig. 4a and Supplementary Figs. 19, 22). This amplifies and also slows down the motions of

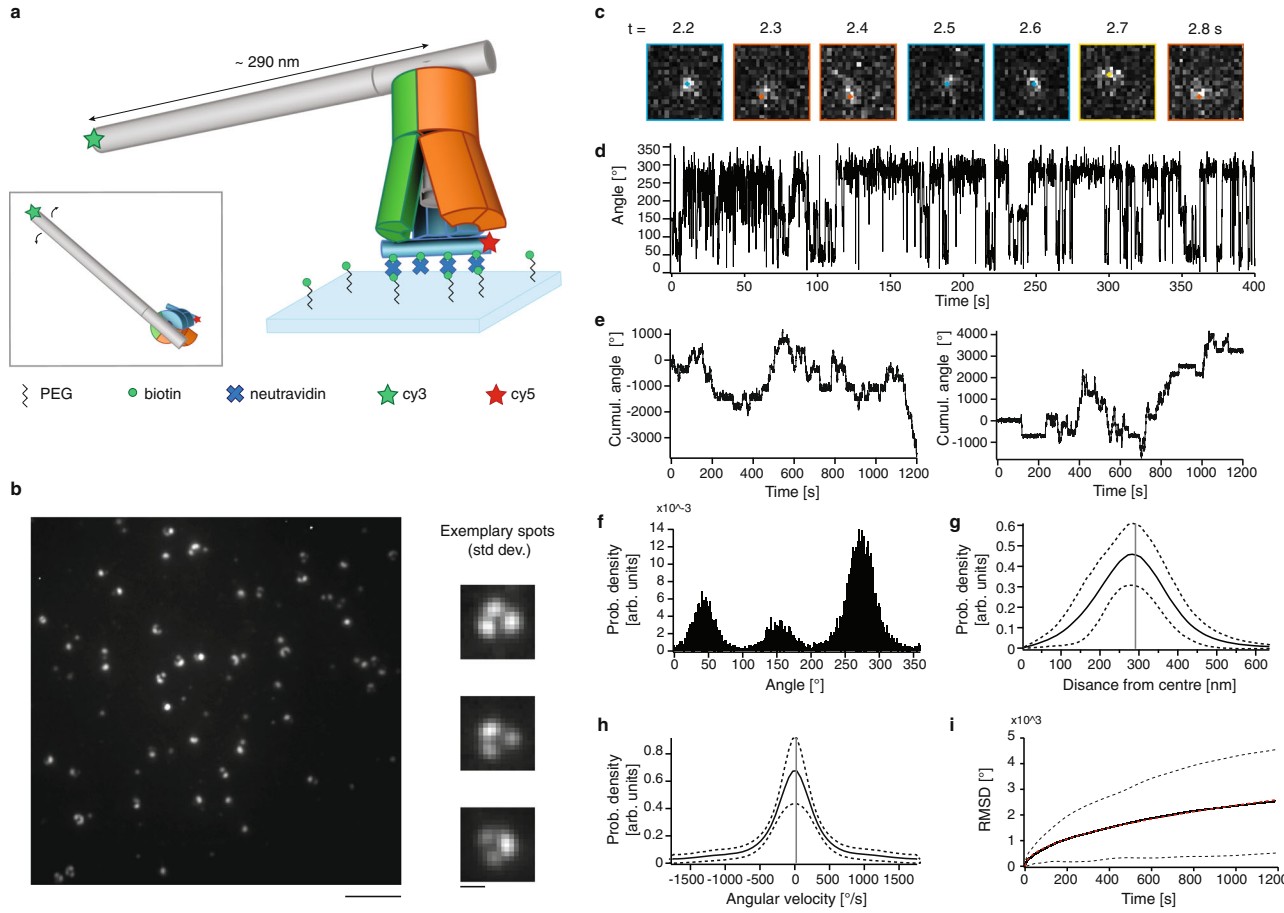

**Fig. 4 Dynamic analysis of the rotary mechanism via single-particle TIRFM. a** Sketch of the experimental set-up: the mechanism is fixed to the cover slide via biotin–neutravidin–biotin bridges. The crossbar of the camshaft is lengthened to approximately 290 nm. Inset: top view. **b** Typical field of view image (left, scale bar: 5 μm) and images of the standard deviation over all frames of single-particle movies for exemplary particles (right, scale bar: 600 nm). **c** Exemplary sequence of frames of a typical particle jumping between three preferred positions (blue, orange, and yellow). **d** Exemplary angle-time trace of a single particle. **e** Cumulative angle over time for two typical particles. **f** Probability density distribution for lever orientations computed from angle-time trace of a typical particle. **g** Solid line: Distribution of the measured distance of lever arm tip from center averaged over $N = 212$ particles. Dashed lines: plus/minus standard deviation. Vertical gray line: weighted average of the distance from center. **h** Solid line: angular velocity distribution computed from $N = 212$ particles. Dashed lines: plus/minus standard deviation. Vertical gray line indicates the weighted average of the angular velocity (which is zero, since there is no directional bias). **i** Solid line: average angular root mean square deviation over time from $N = 212$ particles. Dashed lines: plus/minus standard deviation. Red dashed line: fit using RMSD $= A\sqrt{t}$. Source data are provided as a Source data file.

the camshaft due to friction with the solvent to facilitate tracking the camshaft motions in real-time.

**Rotary random walks.** Imaging of rotary mechanisms with released camshafts revealed particles performing rotary random walks in addition to stationary particles (Fig. 4b). We designed a 160 base pair long extension to the 6hb domain protruding from the stator unit 1 (Supplementary Figs. 23–25) to test whether the stator was well anchored to the surface. We thus obtain two pointers (lever arm and 6hb) that simultaneously track the motions of the camshaft and of the stator, respectively. With this setup, we confirmed that the stator remains fixed (Supplementary Fig. 25b). When the camshaft was fixed to the stator by strand linkages, we observed a negligible fraction of particles exhibiting rotary motion (Supplementary Fig. 25c). We conclude that the rotary particles seen for the sample with released camshaft indeed reflect motions of the camshaft inside the stator.

We used super-resolution centroid tracking[32] (Fig. 4c) to compute single-particle angular orientation trajectories (Fig. 4d). These traces typically featured stepwise jumps between three

different levels. Since the rotation occurs in thermal equilibrium powered by random Brownian motion, no effective directional bias is expected and is also not observed (Fig. 4e). The particles preferentially populate three main orientations separated by 120° (Fig. 4f), which matches with the designed three-fold symmetry of the stator and with the three preferred orientations of the shaft that we saw with cryo-EM. The mean distance of the moving centroid to the center of movement computed for each particle was ~286 nm (Fig. 4g), which corresponds well to the expected 290 nm (Fig. 4a). The angular velocity distribution averaged over all measured rotary particles has an approximately Gaussian shape (Fig. 4h) with speeds in the range of up to 4 revolutions per second. The root mean square deviation (RMSD) of the angular displacements grows with the square root of time in accordance with normal diffusion (Fig. 4i).

**Allosteric coordination of rotor and stator motions.** We designed and self-assembled five additional variants of the stator in which we altered the flexibility of the pawls to see whether the camshaft's rotational diffusion could be influenced (Fig. 5a–f and

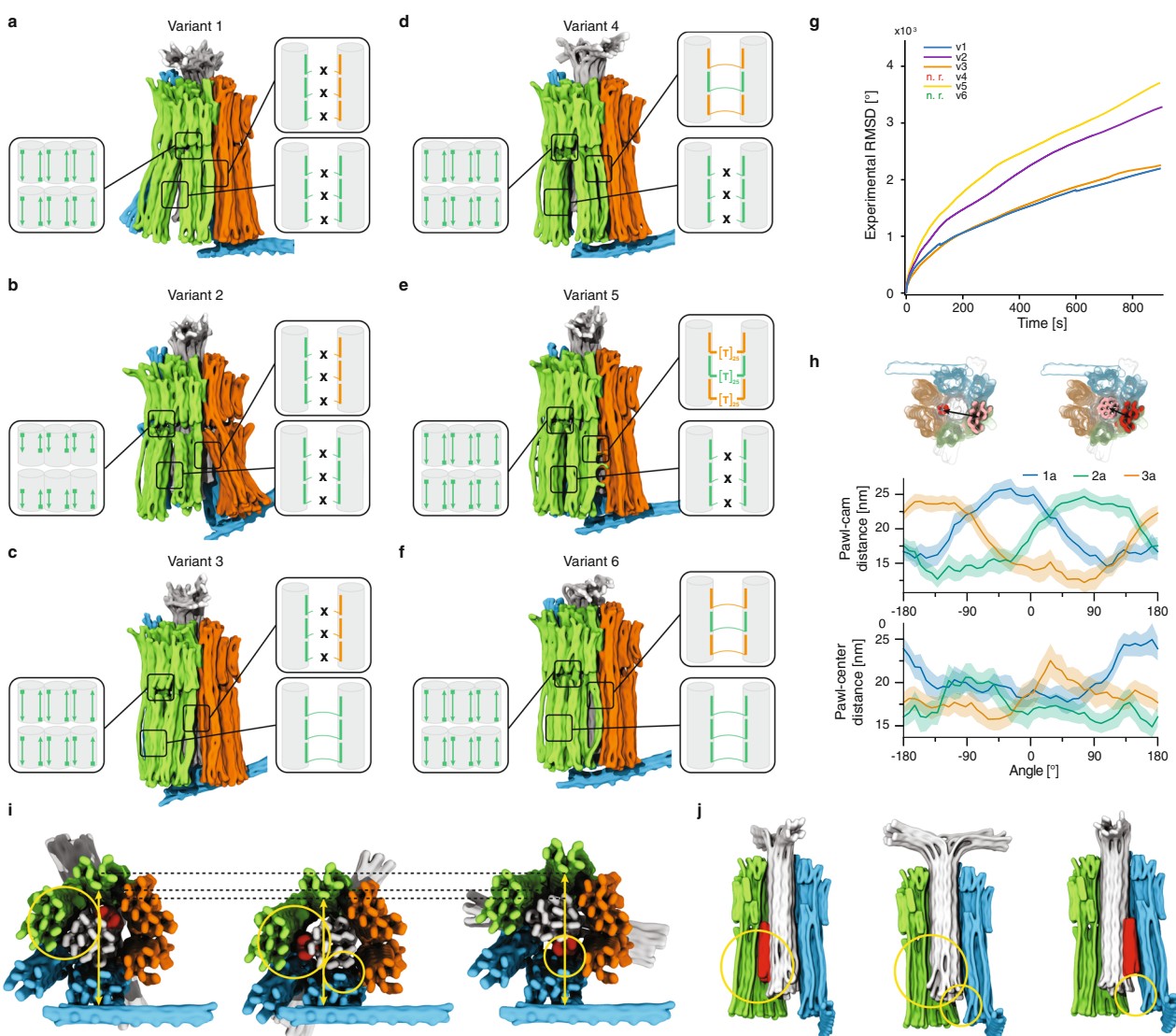

**Fig. 5 Mechanical coupling between stator and camshaft. a–f** Side views of snapshots from 3D structures predicted for variants of the stator using mrDNA, see Supplementary Movie 3. Insets: schematics of design modifications in stator to influence its mechanical properties. Cylinders represent double-stranded DNA helices, while colored lines represent staple strands. **g** Experimentally measured average root mean square deviation for variants 1 (blue), 2 (purple), 3 (orange), and 5 (yellow) from single-particle angle-time traces (see Fig. 4). For variants 4 and 6, no rotation was observed. **h** Molecular dynamics simulation of forced rotation of variant 1. The steered molecular dynamics protocol was applied to a potential acting on the dihedral angle. As the camshaft spins, the cam cyclically approaches each pawl (top, stator unit 1 pawl in blue, stator unit 2 pawl in green, stator unit 3 pawl in orange), causing it to deform away from the center of the camshaft (bottom). The rotation was forced for three full turns in each direction and the results were averaged (thick lines). The shadings represent the standard deviation. **i, j** Average structure from mrDNA simulations of variant 1, extracted from multiple cycles of forced rotation in top (**i**) and side (**j**) views. Big circles: deformations on the stator unit 2; small circles: deformations on stator unit 1. Yellow arrows: deformation of the entire stator during the camshaft rotation. In **j**, the stator unit 3 is not shown. Source data are provided as a Source data file.

Supplementary Figs. 26–31). We released the shaft from its docking site in each sample, acquired single-particle fluorescence microscopy movies, and performed centroid tracking of single rotating particles to compute angular RMSD over time traces (Fig. 5g) as described in Fig. 4. In addition to experiments, these designs were also analyzed dynamically with multi-resolution molecular dynamics (MD) simulations using mrDNA[33] (Supplementary Movie 3).

Variant 1 has no lateral connections between the pawls. Therefore, the pawls can flex independently, as seen in the mrDNA simulations (Fig. 5a). In variant 2, we additionally deactivated the base stacking contacts between bearing and pawls,

which further increases pawl flexibility (Fig. 5b). Experimentally we saw that the rotary mobility of this variant increased approximately by a factor of 2 compared to variant 1 with the stiffer pawls (Fig. 5g). In variant 3, which was already characterized in Fig. 4, we coupled the two pawls within each stator unit laterally with strand crossovers so that they move as one unit (Fig. 5c). This design change removes three of six possible "slots" for the camshaft. Interestingly, variant 3 had very similar rotary mobility compared to variant 1. In variant 4, we coupled the pawls along the lateral interface of neighboring stator units with strand crossovers (Fig. 5d). This design change removes the other three possible slots and had a drastic influence

on mobility: it completely inhibited rotary motion, meaning there was a negligible fraction of rotating particles in this sample (Supplementary Fig. 32a). In variant 5, instead of direct strand crossovers as in variant 4, we used 25-thymidine-long strand linkages. These linkages not only restore pawl flexibility, but they also push the pawls a bit apart due to the volume taken up by the poly-T linkages (Fig. 5e). Strikingly, this design change completely restored rotary mobility (Fig. 5g). In fact, this variant showed the highest diffusive mobility of all variants which we attributed to the increased distance between the pawls. Finally, in variant 6, all the pawls were tightly connected to each other by lateral staple strand crossovers (Fig. 5f). Consistent with the previous results, this variant did not rotate at all (Supplementary Fig. 32b), presumably because the pawls could not give way to the cam and kept it locked in the conformation in which it was docked initially. These observations suggest that the camshaft preferentially populates and switches between the three slots located at the interface between stator units, whereas the slots located in between the two pawls per stator unit are not used.

We used mrDNA simulations for variant 1 to investigate the coupling between the camshaft orientation and the mechanical deformation of the stator (see "Methods"). In these simulations, we enforced the relative orientation of the camshaft and the stator via a harmonic potential that acted on the dihedral angle formed by the centers of geometry of the four regions (Fig. 5h). By changing the rest angle of the potential at a constant rate, the shaft was driven to rotate in each direction for at least three complete revolutions. We analyzed the resulting trajectories by binning and averaging the microscopic configurations according to the camshaft angle every 10°, revealing deformation of each pawl as the cam approached. The distance between the cam of the shaft and each subunit had a roughly sinusoidal dependence on the camshaft angle with an amplitude of ~5 nm and phase offset by ~120° for adjacent subunits, as expected (Fig. 5h, top graph). However, when the cam approached a pawl, it caused the pawl to bend away from the center of the camshaft by ~5 nm as seen by an increase of the distance between each pawl and the center of the shaft (Fig. 5h, bottom row). Similarly, the angle between each pawl and the adjacent pawls was seen to be maximal as the cam approached the outer pawls, and minimal as it approached the central pawl (Supplementary Fig. 33). These deformations can also be seen from exemplary simulation snapshots (Fig. 5i, j): when the cam approaches one of the pawls, they are pushed further from the camshaft center, while they relax back into position if the cam is pointing away. A principal component analysis of the stator (Supplementary Fig. 34 and Supplementary Movie 4) indicated that the dominant motions involve deformation of the pawls on opposite sides of the rotor, revealing three axes that are readily elongated to accommodate the cam.

We generalized the simulation analyses to variants 3 and 6, revealing that all variants exhibit qualitatively similar deformations of the pawls with respect to the cam despite the different inter-pawl connections (Supplementary Fig. 33). However, the deformation is diminished by increased coupling of the pawls as implemented in variant 3 and especially variant 6, which did not show any actual rotation in our experiments. Furthermore, the minimum pawl–shaft distance during the rotation cycle of variant 1 is similar to the maximum distance for variant 6, reflecting that the latter does not readily accommodate the cam. In summary, the simulations show that the rotary motion of the cam is tightly coupled to, and coordinated by, the reciprocal deformation of the pawls, as designed. The coordination and reciprocal motion may be appreciated in movies of the forced rotation simulation results for variants 1, 3, and 6, respectively (Supplementary Movies 5 and 6).

## Discussion

In this work, we presented a compliant nanoscale rotary mechanism with a central camshaft surrounded by a stator with programmable stiffness. We used single-particle cryo-EM to structurally characterize the components, and also the entire mechanism in different states. We also studied the dynamical behavior of the rotary apparatus via TIRFM and molecular dynamic simulations. The results from structural analysis by cryo-EM, the single-particle fluorescence imaging, and the simulations all support the following picture: the camshaft can freely rotate inside the stator, but there exist three preferred shaft orientations. These orientations correspond to states with the cam snapped into slots located at the boundaries between the stator units.

The three preferred states for the cam are defined mechanically, meaning that the camshaft is pressed into the slots by the forces exerted by the surrounding stator. This is a crucial difference to previously reported nanomachines, where conformational states were defined via direct chemical bonds. The mechanical snapping into place now enables regulation at a distance. In our mechanism, the pawls of the surrounding stator must deform to allow rotation of the camshaft and escape from the mechanical slots, as visualized directly by the simulated trajectories. Such deformations occur in our mechanism in a thermally activated fashion, giving rise to Brownian rotary diffusion. Through targeted design changes, we made some versions of the stator less flexible. As a consequence, the rotary movements of the shaft became slower, even stalling the camshaft in two design variants. Together, these experiments demonstrate an allosteric coupling between the orientation of the camshaft and the mechanical deformation of the surrounding bearing and between the rotary motion of the shaft and the reciprocal open/close transitions in the stator.

Our mechanism operates through coordinated motion of its components. As such it could provide a framework for creating artificial nanomachines based on the concept of Brownian ratchets[34,35]. For example, the opening/closing transitions of the pawls in the stator could potentially be gated by the consumption of chemical fuel[6–12] to create a chemically fueled rotary nanomotor. Likewise, due to microscopic reversibility, it is conceivable that such a system could potentially be reversed and used for uphill chemical synthesis, as in ATP synthase, by applying mechanical torque to the central shaft, thus creating a chemical generator. In that pursuit, the coordinated motions of the pawl and the camshaft, or new variants of it, could be employed to cyclically bring reactants into close proximity. All of these applications require the creation of intricately shaped components and their assembly into a functional mechanism. Our work shows a route for how such tasks can be accomplished but also highlights the challenges involved in imparting the desired functionality on such ultraminiaturized molecular mechanisms. We expect that the realization of more complex artificial machinery will go hand in hand with further improvements in analyzing continuous molecular motions by cryo-EM[31,36] and with improved predictive computational approaches[33,37].

## Methods

**Design of the DNA origami nanostructures.** All structures were designed using cadnano0.2[38]. The central shaft was folded from a 7560-bases long scaffold, while the stator units and the lever arm were folded from an 8064-bases long scaffold. The modified stator units 1 for TIRFM measurements were folded from a 9072-bases long scaffold[39]. The 6 helix-bundle bound to the stator unit 1 was folded from a 2873-bases long scaffold[39].

**Folding of the DNA origami nanostructures.** The folding reaction mixtures contained 40 nM scaffold and 160 nM (for structures with p9072 scaffold) or 200 nM (for the other structures) staples (Eurofins MWG, IDT). The folding buffer

included 5 mM TRIS, 1 mM EDTA, 5 mM NaCl and 10–20 mM $MgCl_2$. The folding solutions were thermally annealed using TETRAD (MJ Research, now Biorad) thermal cycling devices. The reactions were left at 65 °C for 15 min and were subsequently subjected to a thermal ramp from 60 to 44 °C (1 °C/h). The folded structures were stored at room temperature (RT). Sequences are reported in Supplementary Data 1-7.

**Purification and concentration of the DNA origami nanostructures.** After folding, all samples were purified via PEG purification, ultrafiltration or physical extraction from agarose gels[30]. For concentrating the monomers, ultrafiltration was used, while ultracentrifugation (110k g, 30 min, RT) was used for concentrating the higher-order structures.

For PEG purification, the DNA origami sample was diluted in a 1:1 ratio with a solution containing 15% PEG 8000. The salt concentration was adjusted to reach a minimum of 10 mM $MgCl_2$. After mixing, the solution was centrifuged at $16k \times g$ for 20 min. The supernatant was removed and the remaining pellet was diluted in a buffer containing 5 mM TRIS, 1 mM EDTA, 5 mM NaCl and 5 mM $MgCl_2$.

For ultrafiltration, Amicon Ultra 0.5 mL or 2 mL Ultracel filters, 50 K and 100 K were used with buffers containing 5 mM TRIS, 1 mM EDTA, 5 mM NaCl and 5 mM $MgCl_2$. An equilibration step with only buffer was performed at $2k \times g$ for 5 min at RT. Then 0.5–2 mL sample (depending on the filter size) was added and centrifuged at $10k \times g$ for 5 min (for the 0.5 mL filters) or $7k \times g$ for 10 min (for the 2 mL filters). Another two rounds of centrifugation were performed adding buffer up to the maximum volume of the filter. To retrieve the sample, the filter was turned upside down in a new tube and centrifuged for 5 min.

For physical extraction from agarose gels, the samples were electrophoresed on 2% gels containing 0.5× TBE and 5.5 mM (for the monomers) or 22 mM (for the higher-order structures) $MgCl_2$ for 1–5 h at 80–90 V. The gels for the monomers were cooled by a water bath, while the gels for the higher-order structures were cooled using a cooling system (Hailea). The desired bands were cut from the gel using an X-tracta Generation 2 hand punch (Biozym). The gel slices were manually squeezed and then centrifuged at $2k \times g$ for 5 min in a Freeze 'N Squeeze DNA Gel Extraction Spin Column with pore size 0.45 μm (BioRad).

**Assembly of the complex.** For the dimerization between stator unit 1 and shaft, a 1:1 solution of the monomers was mixed with 20 mM $MgCl_2$ and left at 40 °C for 16–24 h. The other stator units and the lever arm were added in stoichiometric conditions at 40 mM $MgCl_2$ and left at 40 °C for 1 day/monomer addition. To set the central shaft free to rotate, invader strands were added in excess (2× for cryo-EM analysis, 4× for TIRFM analysis) and the reaction mixture was left in a shaker for 16–24 h at room temperature.

**Agarose gel analysis of the DNA origami nanostructures.** The DNA nanostructures were electrophoresed on 2% agarose gels containing 0.5× TBE and $MgCl_2$ at different concentrations: typically, for monomers and dimers 5.5 mM $MgCl_2$ was used, while 22–33 mM $MgCl_2$ was necessary for higher order structures. For the latter samples, the temperature was kept at around 10 °C with a cooling system (Hailea). The gels were scanned using a Typhoon FLA 9500 laser scanner (GE Healthcare) at a resolution of 50 μm/pixel. For clarity, gel intensities in this paper were auto leveled.

**Negative stain TEM.** Purified structures were adsorbed onto glow-discharge Cu grids with carbon support (in house production and Science Services, Munich) and stained with a 2% aqueous uranyl formate solution containing 25 mM NaOH. Samples were incubated for different time lengths depending on the concentration. In general, structures with concentrations in the order of tens of nM were incubated for 30 s, while lower concentrated samples (5 nM or below) were incubated for 5 to 10 min.

Images were acquired using a Philips CM100 operating at 100 kV or a Tecnai 120 (Thermo Fisher Scientific) at 120 kV. Negative stain 2D class averages were computed using RELION[40] without CTF correction.

**Cryo-EM sample preparation.** For cryo-EM analysis of the monomers, the samples were folded and ultrafiltrated[30] as described in the section "Purification and concentration of the DNA origami nanostructures" until reaching concentrations around 1 μM.

For higher order structures, monomers were folded and gel extracted in order to ensure to have only the right monomeric species. The samples were ultrafiltrated for increasing the concentration. The monomers were polymerized as described in the previous section. The polymers were ultracentrifuged at $110k \times g$ for 30 min at RT. The supernatant was pipetted away leaving only a small volume of sample with the desired high concentration. If needed, the invader strands were added at a 2× excess to binding site for 16-24 h at room temperature.

The samples were incubated onto glow-discharged C-flat 1.2/1.3, C-flat 2/1 or lacey carbon with ultrathin carbon support grids and plunged-frozen using a Vitrobot Mark IV (Thermo Fisher Scientific). The parameters used are given in the table below (SU = stator unit; CS = camshaft; B SU = camshaft bound to stator unit; Free = camshaft set free to rotate; LCS = lacey carbon with ultrathin carbon support).

| Sample | SU1 | SU2 | SU3 | CS | Stator | B SU1 | B SU2 | B SU3 | Free |
|---|---|---|---|---|---|---|---|---|---|
| Volume [μL] | 4 | 4 | 4 | 3 | 4 | 4 | 3–4 | 3–4 | 4 |
| Conc. [nM] | 1800 | 1400 | 780 | 129 | 80 | 3–100 | 40–85 | 65–92 | 35–93 |
| Humidity [%] | 100 | 100 | 100 | 100 | 100 | 100 | 100 | 100 | 100 |
| Temp. [°C] | 20 | 20 | 20 | 20 | 20 | 20 | 20 | 20 | 20 |
| Wait time [s] | 0 | 0 | 0 | 60 | 0-5 | 0–600 | 0 | 0 | 0–15 |
| Blot time [s] | 1 | 2 | 2 | 0 | 1–2 | 0–3 | 0–2 | 0–2 | 0–3 |
| Blot force [mm] | 0 | 0 | 0 | 1 | 1–2 | 0–1 | 0–2 | 1–2 | 1–2 |
| Drain time [s] | 0 | 0 | 0 | 3 | 0 | 0 | 0 | 0 | 0 |
| Grid type | C-flat 1.2/1.3 | C-flat 1.2/1.3 | C-flat 1.2/1.3 | C-flat 1.2/1.3 | C-flat 1.2/1.3 | C-flat 1.2/1.3, LCS | C-flat 1.2/1.3 | C-flat 1.2/1.3 | C-flat 1.2/1.3 |
| Number of grids | 1 | 1 | 1 | 1 | 2 | 9 | 6 | 5 | 8 |

**Cryo-EM image acquisition.** The camshaft was manually imaged using a Tecnai 120 (Thermo Fisher Scientific). All the other samples were imaged automatically with a Titan Krios (Thermo Fisher Scientific). Imaging parameters are listed in the table below.

| Sample | SU1 | SU2 | SU3 | CS | Stator | B SU1 | B SU2 | B SU3 | Free |
|---|---|---|---|---|---|---|---|---|---|
| Microscope | Titan | Titan | Titan | Tecnai 120 | Titan | Titan | Titan | Titan | Titan |
| Voltage [kV] | 300 | 300 | 300 | 120 | 300 | 300 | 300 | 300 | 300 |
| Magnification | ×29k | ×29k | ×29k | ×30k | ×29k | ×37k | ×29k | ×29k | ×29k |
| Spot size | 4 | 4 | 4 | 4 | 4 | 5 | 4 | 4 | 4 |
| Defocus [μm] | −2 | −2 | −2 | −2 | −2 | −2 | −2 | −2 | −2 |
| Dose [$e^-/A^2$] | ~40 | ~40 | ~40 | / | ~40 | ~45 | ~50 | ~40 | ~50 |
| Exposure time [s] | 2.6 | 2.6 | 2.6 | / | 2.6 | 1.5 | 2.6 | 2.6 | 2.6 |
| Pixel size [A] | 2.32 | 2.32 | 2.32 | 3.37 | 2.32 | 1.82 | 2.32 | 2.32 | 2.32 |
| Frames | 105 | 105 | 105 | / | 105 | 63 | 105 | 105 | 105 |
| Fraction | 7 | 7 | 7 | / | 7 | 9 | 7 | 7 | 10 |

**Cryo-EM image processing.** The movies acquired with the Titan were subjected to motion correction using MotionCor2[41]. All the micrographs were CTF corrected with CTFFIND4[42]. The camshaft particles were picked manually, while for the other samples particles were picked with RELION[40] or with crYOLO[43]. Other processing steps were performed with RELION (RELION 2 for the central camshaft, RELION 3.0 for the other structures). Multiple runs of 2D and 3D classifications were typically performed to exclude incompletely folded particles or particles lying on the carbon film. The 3D classes presenting most features were further refined and post-processed.

For the complex set free to rotate, a 3D classification without alignment was performed after refinement, using multiple maps as references, i.e., the three reconstructions of the structure with the camshaft fixed to the stator units. With this method, three different positions of the central camshaft could be found.

The stator units were further subjected to a multibody refinement[31], where each of the pawls and the bearing were treated as a separate body. Similarly, the higher order structures were subjected to multibody refinements where each of the monomers composing the complex was considered a separate rigid body.

For the analysis of the pawls' motion for the sample with the released camshaft, all the particles in the final three 3D classes showing different conformations were considered. Ten rigid bodies were defined: the camshaft, the 6 pawls (2 per each stator unit) and the 3 bearings (1 per each stator unit). Multibody and principle component analysis were used to identify the main motions. Two main pawls' motions could be determined. The extreme maps were then overlapped in UCSF Chimera[44]. One helix per pawl in the first extreme map was identified and the distance to the same helix in the second extreme map was calculated with UCSF Chimera.

**Assembly for TIRFM.** Monomers were folded and purified using physical gel extraction. The samples were ultrafiltrated for increasing the concentration. The monomers were polymerized as described before. If needed, the invader was added at a 4× excess to binding site for 16-24 h at room temperature. Biotinylated oligos were incubated with a 32× excess neutravidin and then added to the polymers in an 8× excess to binding site for 1–2 h at RT. The resulting reaction mixture was gel purified for extracting only the correct species. Sample concentrations varied between 100 and 700 pM. Samples were left at RT for at most 2 days and then imaged at the microscope.

**TIRFM movie acquisition.** Microscope cover slides (Sigma Aldrich) were cleaned in 2 M NaOH for 30 min, rinsed with $ddH_2O$ and then sonicated for 5 min in a 2% Hellmanex solution. The slides were rinsed in $ddH_2O$, sonicated in $ddH_2O$ for 5 min, rinsed again and then sonicated in ethanol (99%). The slides were dried at 70 °C for 1 h. A solution of 0.5% bioPEG-silane solved in ethanol with 1% acetic acid was incubated on the slides at 70 °C for 30 min. The slides were then rinsed with $ddH_2O$, dried with $N_2$, and stored protected from light[45].

An acrylic glass template containing 4 chambers was sealed to the cover slide with vacuum grease. The chambers were washed with a buffer containing 10 mM TRIS, 2 mM EDTA, 10 mM NaCl, 40 mM $MgCl_2$ for 3-4 times. 50 μL sample was

incubated for 5 to 20 min depending on the sample concentration. The chamber was washed again 3–5 times with a buffer containing 5 mM TRIS, 1 mM EDTA and 500 to 1000 mM NaCl. Afterwards, the chamber was washed 3 times with imaging buffer containing an oxygen scavenging system (50 mM TRIS pH 8, 1 mM EDTA, 500 mM NaCl, 2 mM Trolox, 0.8% D-glucose, 442 U/ml glucose oxidase, 2170 U/ml catalase) prior to data acquisition. Enzymes, Trolox and glucose were purchased from Sigma Aldrich. Movies were acquired at room temperature with a custom-built objective-type TIRFM[15,45]. Movies were acquired for 2–20 min at a frame rate of 20 frames/s and a laser on time of 5 ms.

**TIRFM movie processing**. Movies were drift-corrected using a FIJI plugin (NanoJ-Core[46]). All successive steps were performed using a custom MATLAB (R2019b) script. Moving particles were manually picked and their frame-by-frame standard deviation was computed. Defective particles or particles showing no motion were sorted out by inspecting the standard deviation images. The remaining particles were further processed by tracking the position of the lever arm in each frame using a virtual window center of mass approach (VWCM). The obtained spots were clustered in 3 groups indicating the three preferred positions of the central camshaft. From the tracked spots, angular positions, RMSD and angular velocities ($\Omega$) were calculated according to:

$$\text{RMSD} = \sqrt{\frac{\sum_{i=1}^{N}(\theta_i - \theta_0)^2}{N}} \qquad (1)$$

$$\Omega_i = \frac{\theta_i - \theta_{i-1}}{\Delta t} \qquad (2)$$

where $N$ is the number of frames, $i$ indicates the $i$-th frame, $\theta_0$ is the angle at $t = 0$, and $\Delta t$ indicates the time difference between 2 consecutive frames. A histogram of the angular velocities was calculated.

**Multi-resolution simulations**. Each variant was simulated using the mrDNA multi-resolution modeling framework[33], first at a resolution of ~5 bp/bead with a 200 fs timestep for 20 μs, followed by simulations at 2-bp/bead resolution and a 40-fs timestep that lasted 80 ns. The higher resolution simulations introduced a local representation of the orientation of the major groove that facilitated construction of an atomic model of each variant. The temperature was held at 295 K. The computational model of DNA–DNA interactions was previously parameterized[33] to match the experimentally measured osmotic pressure in a DNA condensate at 20 mM MgCl₂ electrolyte. The stacking interactions between the DNA helices in the bearing region and the pawls were modeled within the mrDNA framework using a custom script that effectively made the two helices contributing to the stacking site a single continuous dsDNA helix, with the azimuthal angle of the helices on both sides of each stacking site being in phase.

Selected variants (1, 3, and 6) were additionally simulated with an applied bias to drive the rotation. The simulations were performed as described above, except a harmonic potential ($k_{\text{spring}} = 0.5$ kcal/mol/degree$^2$) was placed on a dihedral angle formed by the centers of geometry of the following four groups of particles: (i) beads constituting the central 60 bp of each of the six-helices that formed the "cam" of the rotor; (ii) beads in the same 60-bp-thick plane of the caDNAno design that formed the 24-helix shaft of the rotor; (iii) a 60-bp-thick slab of beads in the shaft of the rotor near the bearing region separated by 85 bp (center-to-center) from the other group of beads on the shaft; and (iv) a 40-bp thick slab of beads in the center of the second subunit of the stator (between 50 and 90 bp from the bottom edge of the bearing). For each variant, two simulations were performed with the rest angle of the harmonic potential increasing in one and decreasing in the other. The simulations were performed for a sufficiently long period of time to observe three complete rotations in each direction.

## Data availability
The cryo-EM maps that support the findings of this study are available in the Electron Microscopy Data Bank (EMDB) under the accession codes EMD-13565, EMD-13566, EMD-13567, EMD-13568, EMD-13569, EMD-13570, EMD-13571, EMD-13572, EMD-13573. See Supplementary Table 1 for identification of maps and EMBD codes. Cryo-EM and real-time fluorescence movie raw data are available from the corresponding author upon request. Sequences of oligos and scaffolds are available in Supplementary Data Files 1–7. Source data are provided with this paper.

## Code availability
The MATLAB scripts for TIRFM data analysis can be downloaded at https://github.com/DietzlabTUM/matlab_tirfm_movies[47]. The script used to set up the SMD simulation is available as Supplementary Software. The simulation trajectories and analysis scripts are available upon request.

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

## Acknowledgements

This work was supported by a European Research Council Consolidator Grant to H.D. (GA no. 724261), the Deutsche Forschungsgemeinschaft through grants provided within the Gottfried-Wilhelm-Leibniz Program, and the SFB863 TPA9 Project ID 111166240 (to H.D.). A.A. and C.M.M. acknowledge support through National Science Foundation (USA) under grant DMR-1827346 and the National Institutes of Health under grant P41-GM104601 (to A.A.). Supercomputer time was provided through Leadership Resource Allocation MCB20012 on Frontera. We thank Massimo Kube and Dr. Fabian Kohler for helpful discussions on the cryo-EM reconstructions, Anna-Katharina Pumm and Dr. Wouter Engelen for support with the MATLAB script and the fluorescence microscopy experiments, and Alexander Koch for auxiliary experiments.

## Author contributions

H.D. designed the research. E.B. and T.D. performed the research. M.N.H. provided the custom scaffold. C.M.M. and A.A. performed the simulations. E.B., C.M.M., H.D., and A.A. prepared the figures and wrote the manuscript. All authors have given approval of the final version of the manuscript.

## Funding

## Competing interests

The authors declare no competing interests.
