## [Peer Review File · Nature Communications]

A nanoscale reciprocating rotary mechanism with coordinated mobility controlREVIEWER COMMENTS

Reviewer #1 (Remarks to the Author):

In this well-written study, Bertosin et al. designed a DNA origami based nanorotor with a central camshaft surrounded by a stator with programmable stiffness. The authors used single-particle cryo-electron microscopy to structurally characterize the rotary apparatus. They further studied the Brownian dynamical behavior of the rotary apparatus via total internal reflection fluorescence (TIRF) microscopy and molecular dynamic simulations and showed that the camshaft can freely rotate inside the stator with three preferred shaft orientations. While the development of nanomotors is an important goal in nanotechnology, I think that the lack of novelty compared to similar published works is a significant shortcoming of this study.

As discussed in their manuscript, there are already similar DNA origami-based molecule rotors and machines such as a tight-fitting passive DNA origami rotary apparatus (by the corresponding author ref. 15) and a robotic arm controlled by an electric field (ref 23). However, the most relevant publication concerning the current work is a report by Ahmadi et al. in Small (2020) in which also the Brownian motion and the flow-driven rotational dynamics of a DNA origami rotor comprising also four components was studied in a microfluidic chamber set-up via TIRF microscopy, showing that the camshaft/propeller can rapidly switch between three preferred rotational states.

Other aspects that the authors should consider are:

195-202: The (non-rotating) controls in the single-particle fluorescence microscopy are poorly described and unclear. Further, the data from these experiments is missing. These controls are important to understand the rotation and flexibility of the whole structure and the lever arm.

203-215: Based on Fig4, the nanostructure was designed to enable a very slow rotational diffusion. While this feature is very useful regarding the single-particle TIRF analysis, it gives the impression that the structure is rather a static nanostructure with structure-switching capacity instead of a real nanorotor. The discussed ATPase has orders of magnitude higher rotational speed, and it remains unclear whether this nanostructure could be useful for future nanomachineries.

219-239, Fig 5G: The non-rotating variants are good controls and should be also analysed. The RMSD values resulting from the Brownian motion of the lever arm should be also plotted to allow an in vitro experimental comparison of all variants.

217-261: For the MD simulations, a trajectory analysis (ideally of the whole rotor) is essential. There, based on the in vitro results, we would expect three dominant states.

At this point, this work does not add much to the state of art in the field especially regarding novelty. Therefore, I am not convinced that this work breaks sufficient new ground to make it suitable to be published in nature communications.

Reviewer #2 (Remarks to the Author):

The manuscript by Dietz and coworkers describes the design and construction of a rotor machine made of DNA synthetic strands assembled through a multilayer DNA origami approach. The rotor is characterized through beautiful Cryo-EM experiments and fluorescence imaging. The paper is very interesting, novel, well-written and well organized. The figures (as usual with Dietz's works) are really well done and clear.

I have no hesitation to support publication of the manuscript in Nature Communications.

Before that I have some "minor" comments and suggestions that I would like the authors to take care for a possible improvement of the clarity of the results and of their future impact in the field:

1) Although figure 1 is really nice I had some hard time to follow some of the features of the rotor. For example, it is not clear how the stator units go from open to closed conformation and what is the function of this conformational change. Also, the Assembly and Release panels (F and G) are not too clear. If I understood correctly the strand displacement reaction used in these two steps is only needed for the assembly of the entire machine. But, what is missing in this figure is the function of this machine! Figure 1 is showing very static units but how the machine is made free to move and what is the movement of the machine is not shown. As this is a "summary" figure showing the idea of the manuscript I think it is worth adding an extra panel with the expected movement of the rotor and how this is induced (how one goes from the static rotor to the moving rotor). I guess this was probably intended to be panel A (the legend says that this panel shows the sketches of the mechanism) but to me this is not super clear. Also, the arrows in panel A suggests a preferred direction of the movement but I don't think this is the case.

2) Cryo-EM images are beautiful. Only comment: is there a way to have more "quantitative" information from these images? For example, when the authors say "We analyzed the data set first using 3D classification, which revealed three dominant, structurally distinct 3D classes in the data set (Fig. 3D)." what does this really means? How "dominant" are these three classes? Can the authors define a quantitative value to understand how many of the rotors are in the 3 conformations? Can the authors define what is the statistical significance of these values? This is more a curiosity from my side as I am quite ignorant about Cryo-EM results analysis...

3) Now, the more interesting part is obviously the one regarding the characterization of the rotor movement. This is done with fluorophores in different positions. The authors also extended the T-crossbar of the camshaft with a 10-helix bundle lever arm resulting in a ~ 290 nm long "pointer". Authors suggest that this "amplifies and also slows down the motions of the camshaft due to friction with the solvent to facilitate tracking the camshaft motions in real-time". Can the authors elucidate this? What is the effect of this elongation on the rotor movement speed? According to the sentence reported above one would expect that the speed of the rotor could be modulated at will by using longer or shorter arms...can the authors explain this better in the text and report some demonstration of this? For example, do the authors have attempted the same fluorescence experiments without an elongated arm?

4) From what I understand there is no preferred orientation of rotation (Brownian diffusion without directional bias). It would be nice to understand if this is totally expected or if there could be some bias due, for example to the composition of the rotor (DNA sequences) or other factors. Also, could a direction be designed on purpose? This could be interesting.

5) From Figure 4D and 4F it appears that one conformation is preferred over the others. Actually the three seem to have all different statistical distributions. Can the authors explain this? Could it be that the interaction between Cy3 in the arm and Cy5 (in a specific stator) affects the statistical distribution of the rotation? Or this is due to other reasons?

6) "Particles can apparently rotate with speeds in the range of up to 3 revolutions per second". Can the authors clarify this sentence? How was this calculated? This also refers back to my comment n. 3.

7) In more general terms the rotor cannot be really defined as a machine: the motion is brownian and thus really the contrary of a machine (maybe I am too drastic in this...). Can the authors comment on this in the conclusions?

8) The authors also say that "it could be of interest to direct the opening/closing transitions in the stator by the consumption of chemical fuel (6-12) to create a chemically fueled rotary nanomotor". However, this should also be coupled with a preferred direction of the rotor. Maybe this should be discussed by the authors (see previous point).

9) Also, and more importantly in my opinion, another aspect should be discussed in the conclusions. Living machines often (almost always) rely on non-equilibrium processes fueled by chemical energy. This makes it possible to perform work and repeated tasks. A continuous supply of chemical energy is required to maintain biological machines in an active state, giving rise to life-like properties such as growth, motion, communication, and adaptation (some of these functions have been described in the introduction). The utilization of energy-dissipating mechanisms, could bring the field of DNA nanotechnology and of DNA machines closer to building "living" devices and materials. It would be good to discuss this in the conclusions. While different dissipative DNA switches and DNA structures have been described recently, dissipative DNA origami machines have yet to be described (I think)!

Reviewer #3 (Remarks to the Author):

In this impressive work titled "A nanoscale reciprocating rotary mechanism with coordinated mobility control", inspired by biological ATP-synthase rotary motor, Bertosin et al. developed a passive rotary device made of DNA origami, demonstrating, to my opinion, the most intricate rationally designed DNA origami dynamical structure fabricated thus far, and significantly advanced several key tools and features necessary for the development of DNA nanoengineering, a field that has already radically transform our ability to fabricate and control bioinspired nanomachines and devices.

Specifically, using clever blunt ends interactions, a method invented previously by the group, the authors combined four different origami constructs (six in some cases) to create a passive rotary device that demonstrate long distance allosteric coordination (dozens of nanometers) between a rotating shaft and three stationary pawls that help holding the shaft in place, resembling the allosteric coordination observed in ATP-synthase. Allosteric coordination enables mechanical transfer of information across different sections of a given molecular complex, and it is therefore essential for the proper function of many biological machines and enzymes in which different sections of the molecular complex that perform different tasks needs to operate in coordination. The authors were able to demonstrate that rational design of DNA origami structures according to engineering rules and experimental and computational practices developed by the DNA nanotechnology community in recent years, many of which by the authors research groups, can reproduce such intricate large-distance coordination in DNA based synthetic complexes.

The long-distance allosteric control was demonstrated by showing that changes in the type and strength of a set of orthogonal interactions between pawls and between them and the bearing element (but not with the shaft), can controlled the shaft rotations speed, thus, information about the dynamics of one part of the complex influence the behavior of another part in a rational, controlled and understood way and (generally) as designed, to my opinion, a major achievement. Furthermore, the authors smart usage of Cryo-EM and single-molecule fluorescence to monitor the fabrication of the rotary complex and to resolve and understand its complex structural and dynamical behavior (together with the usage of MrDNA computer simulations), is an excellent demonstration of how to further advance the field.

The work contains several design and experimental elements that are worth mentioning. The design complex is probably the most tight-fit origami structure demonstrated thus far, and it contains more intricate different structural elements (such as a rotating shaft and its bulging helical element that prevents shaft dissociation, three different moving pawls, a hollow bearing, several blunt-ended helical elements that lock the origami complex together, a 290 nm pointer) than any other DNA complex I am familiar with. Successfully putting all these elements together is impressive. Further, the excellent usage of Cryo-EM and single-molecule fluorescence imaging to resolve the rotary motor complex structure and dynamics. With these tools the authors monitored the rotor fabrication process and characterize the rotor dynamics including shaft states and the shaft angle and rotation rates distributions and demonstrated the expected Brownian dynamics. Taken together, I consider the above list a major achievement that significantly promote the field on several fronts.

Finally, I completely agree with the authors suggestion that the passive rotary device and the allosteric function presented here are major steps towards effective nonautonomous, and finally, autonomous chemical-energy driven rotary motors, a goal considered a 'holy grail' in the field of artificial molecular machines.

The analysis of the results seems correct, and the manuscript and the figures are generally clear (see the 'minor comments' section below).

For the reasons detailed above I consider this an excellent work and I am sure that it will be interesting for both experts and more general audience alike and that it should be published in Nature Communications.

Below are two minor comments that may help improve the manuscript.

Minor comments:

1. This is a minor comment, but I am not sure that the velocity distribution should behave in a gaussian fashion (the angle distribution should indeed be gaussian), rather I think it should behave as two exponents (one for the negative and the other for the positive velocities), and the seemingly gaussian distribution observed is a result of the method inherent limitation to resolve the high frequency of low-velocities events. In other words, the experimental/instrumental response function (IRF) smooths the exponents to look somewhat like a gaussian in low velocities, and the long tails towards high positive and negative velocities are explained by the two exponents long tails. If this is correct the experimental results fit even better to the theory, demonstrating even higher data quality. Please check.
2. In some places the text is not immediately clear and is too long (mainly in the results section). Some more editing may improve the manuscript.

Excellent work.

Reviewer #4 (Remarks to the Author):

The manuscript describes the design and fabrication of DNA structures that can assemble to form a rotary complex. The authors point to inspiration from rotary ATPases such as the ATP synthase as biological examples of remarkable molecular motors. Similar to ATP synthase, the authors design a three part stator and a rotor that can rotate inside the stator complex. However, unlike the ATP synthase, the synthetic complex does not couple rotary motion to chemical or mechanical work. Despite so, the work provides an important step towards being able to make nanoscale molecular motors and the authors characterized their designs using a variety of biophysical tools.

1. The authors do not provide quantification of the amount of structural deformation observed during rotation of the rotor. How much are the pawls displaced as the rotor rotates? Is this deformation different from the displacement observed during simple Brownian motion of the pawls? This can probably be estimated from the cryo-EM structures, and will provide an indication of how well chemical and mechanical work can be coupled to rotary motion as the authors indicate in the discussion.
2. The entire rotary complex is very flexible, as indicated by the cryo-EM data and depicted in the videos. In the most flexible variants, the pawls seem almost randomly displaced. The authors show that flexibility of the pawls can be tuned to increase stiffness, but that also seems to prevent mechanical displacement of the pawls. Is there a middle ground that allows the pawls to remain stationary in the ground state, but be flexible enough to be displaced during rotary action? Additional plots/figures showing stiffness to pawl displacement might help to tease this out.
3. While the authors do not show in this study how chemical and mechanical work could be precisely coupled to rotary motion, this point should be discussed in more detail in the discussion section.

Minor points:

- Structures in figures need scale bars.
- Figure 2 – Panel G is misleading, it seems like the structure of the stator+rotor complex; consider removing panels
- Figure 2H is difficult to interpret
- Figure 3 – how is rotation locked vs released? Described in text but not clear how it works

REPLY TO REVIEWERS

We thank the reviewers for their time and their effort spent in reviewing our work. We appreciate the constructive comments and criticism. We have addressed all comments in our point-by-point response below.

The reviewer comments are given in **bold face**, the author responses are in plain text, while revised manuscript text are quoted in times new roman.

Reviewer #1 (Remarks to the Author):

In this well-written study, Bertosin et al. designed a DNA origami based nanorotor with a central camshaft surrounded by a stator with programmable stiffness. The authors used single-particle cryo-electron microscopy to structurally characterize the rotary apparatus. They further studied the Brownian dynamical behavior of the rotary apparatus via total internal reflection fluorescence (TIRF) microscopy and molecular dynamic simulations and showed that the camshaft can freely rotate inside the stator with three preferred shaft orientations. While the development of nanomotors is an important goal in nanotechnology, I think that the lack of novelty compared to similar published works is a significant shortcoming of this study.

As discussed in their manuscript, there are already similar DNA origami-based molecule rotors and machines such as a tight-fitting passive DNA origami rotary apparatus (by the corresponding author ref. 15) and a robotic arm controlled by an electric field (ref 23). However, the most relevant publication concerning the current work is a report by Ahmadi et al. in Small (2020) in which also the Brownian motion and the flow-driven rotational dynamics of a DNA origami rotor comprising also four components was studied in a microfluidic chamber set-up via TIRF microscopy, showing that the camshaft/propeller can rapidly switch between three preferred rotational states.

The reviewer expresses some concerns regarding the novelty of our work, on the grounds of our system moving between three preferred states. However, the switching between three states is not the key the point of our work. Our nanostructure is capable of mechanically transmitting motions between its components, and this is the novel aspect. Furthermore, the present work is not about absolute size – it is about the dynamic interaction between movable components.

There are many conceptual and technical advances presented in our current work, which have no precedent in the literature: in our complex we implemented mechanically coupled motions between rotor (camshaft) and stator, in a similar way to motor components in the macroscopic world. Also, the rotation is related to the conformational changes of the stator (similar to working mechanism of protein and enzymes). Indeed, the stator variant that was completely rigid suppressed the rotation of the rotor, meaning that the conformational changes of the stator are absolutely necessary for the rotor to rotate.

Moreover, we provide in depth and high-resolution single-molecule fluorescence and cryo-electron microscopy data, not only localizing the position of the camshaft in time but also calculating important dynamics observables for the rotational behavior, to derive a convincing picture of the actual motor dynamics.

Other aspects that the authors should consider are:

195-202: The (non-rotating) controls in the single-particle fluorescence microscopy are

poorly described and unclear. Further, the data from these experiments is missing. These controls are important to understand the rotation and flexibility of the whole structure and the lever arm.

To address this comment, we added the following figure into the supplementary information:

Figure S25 Control measurements on TIRFM. (A) Schematic representation of the complex with the extended pointer. (B) Typical field of view image (top, scale bar: 5 μ m) and single-particle standard deviation images (bottom, scale bar: 600 nm) in the cy5 channel (stator) for the sample with the extended pointer. (C) Typical field of view image (top, scale bar: 5 μ m) and images showing the standard deviation over all frames of single-particle movies (bottom, scale bar: 600 nm) in the cy3 channel (lever arm) for the sample with the camshaft bound to the stator unit 1. (D) Images showing the standard deviation over all frames of single-particle movies of particles switching between 3 spots. Scale bar: 600 nm

Panel A shows the set-up for the control measurements. We added a cy5 dye on the tip of the elongated 6hb. In this way, if the whole complex (and not only the camshaft) should rotate, we would be able to detect it. This is displayed in panel B. The spots are not showing any rotational movement, meaning that the stator is anchored to the glass slide.

Panel C is a field of view in the cy3 channel of another sample, where the camshaft was bound to the stator unit 1. The particles display no rotational motion. These two datasets demonstrate that the rotating particles in fig 4 of the manuscript really represents rotating camshafts and no artefacts. To highlight the difference, panel D shows images of the standard deviation over all frames of single-particle movies of particles switching between 3 spots (see also Fig. 4 in manuscript).

203-215: Based on Fig4, the nanostructure was designed to enable a very slow rotational diffusion. While this feature is very useful regarding the single-particle TIRF analysis, it gives the impression that the structure is rather a static nanostructure with structure-switching capacity instead of a real nanorotor.

We note that the speed is not slow at all, with angular velocities up to ~1500°/s. Also, we artificially slowed the rotor to facilitate detection down by the friction of the elongated long lever arm. Furthermore, a nanostructure which has structure-switching capacity is obviously not static. We finally note that the FoF1 ATP synthase fundamentally operates via a structure-switching mechanism driven by ATP binding, hydrolysis, and release. Nevertheless, the ATPase is widely regarded to be an exemplary biological rotor.

The discussed ATPase has orders of magnitude higher rotational speed, and it remains unclear whether this nanostructure could be useful for future nanomachineries.

We note that the angular velocity of ATPase depends on ATP concentration, at low ATP concentration its angular velocity becomes in fact similar to our object. The usefulness of our

nanomachine is not related to its speed (and it is in fact among the fastest artificial rotary systems developed thus far). It is derived from the fact that we were able to couple the motion of the stator to the rotation of the rotor, thus having a coordinated motion.

219-239, Fig 5G: The non-rotating variants are good controls and should be also analysed. The RMSD values resulting from the Brownian motion of the lever arm should be also plotted to allow an in vitro experimental comparison of all variants.

We added typical field of view images in the supplementary information file to show that v4 and v6 are not rotating.

Figure S32 Suppression of the rotation in v4 and v6. (A) Typical field of view (left, scale bar: 5 μm) and single-particle standard deviation images (right, scale bar: 600 nm) of v4 in the cy3 channel. (B) Typical field of view (left, scale bar: 5 μm) and images of the standard deviation over all frames of single-particle movies (right, scale bar: 600 nm) of v6 in the cy3 channel.

Panel A shows a typical field of view of v4 while panel B shows a typical field of view of v6 in the cy3 channel (lever arm). Moreover, we calculated the RMSD of v4 and v6 and compared it with the RMSD of the other variants (compare to Fig. 5 in manuscript). As shown in the Figure below, the RMSD for v4 and v6 are near 0, meaning that these variants are stationary.

Figure 1 RMSD computed for all the variants.

217-261: For the MD simulations, a trajectory analysis (ideally of the whole rotor) is essential. There, based on the in vitro results, we would expect three dominant states.

We performed a whole trajectory principal components analysis of the stator, which revealed that the dominant motions result in gaps that could accommodate the cam with pseudo-three-fold symmetry, consistent with the experimental observations. A figure depicting the dominant PCA modes is included as Fig. S34 in the Supplementary Information and is reproduced below, and a sentence added to the main text briefly describes the new analysis. Supplementary Movie 6, which depicts the principal modes shown here, was also added.

Figure S2: Principal components analysis of pawl configurations. PCA was performed using the scikit-cuda PCA algorithm to extract the top 100 eigenvectors of the cartesian coordinate covariance matrix of each variant 1, 3, and 6 during forced rotation and after aligning the base of the stator. The camshaft was included in the simulations, but excluded from the PCA analysis. The top five modes, accounting for ~50% of the total variance, are visualized here as a deviation (subunits colored blue, orange and green) from the mean structure (white) depicted alone in the first column. The amplitude of the PCA mode deviations was enhanced by the same factor for all modes to make the motion more discernable. Red arrows highlight sites that open up in the PCA mode and are likely to accommodate the rotor cam.

At this point, this work does not add much to the state of art in the field especially regarding novelty. Therefore, I am not convinced that this work breaks sufficient new ground to make it suitable to be published in nature communications.

As stated above, the novelty of this work is about the rotation of the camshaft being tightly coupled to the conformational changes of the stator which is an important milestone toward completely autonomous artificial molecular motor in the future. The fact that we are able to couple together different motions through conformational changes brings us nearer to creating a DNA origami motor similar to the natural ones. We find this criticism thus unwarranted.

Reviewer #2 (Remarks to the Author):

The manuscript by Dietz and coworkers describes the design and construction of a rotor machine made of DNA synthetic strands assembled through a multilayer DNA origami approach. The rotor is characterized through beautiful Cryo-EM experiments and fluorescence imaging. The paper is very interesting, novel, well-written and well organized. The figures (as usual with Dietz's works) are really well done and clear.

I have no hesitation to support publication of the manuscript in Nature Communications.

Before that I have some “minor” comments and suggestions that I would like the authors to take care for a possible improvement of the clarity of the results and of their future impact in the field:

1) Although figure 1 is really nice I had some hard time to follow some of the features of the rotor. For example, it is not clear how the stator units go from open to closed conformation and what is the function of this conformational change. Also, the Assembly and Release panels (F and G) are not too clear. If I understood correctly the strand displacement reaction used in these two steps is only needed for the assembly of the entire machine. But, what is missing in this figure is the function of this machine! Figure 1 is showing very static units but how the machine is made free to move and what is the movement of the machine is not shown. As this is a “summary” figure showing the idea of the manuscript I think it is worth adding an extra panel with the expected movement of the rotor and how this is induced (how one goes from the static rotor to the moving rotor). I guess this was probably intended to be panel A (the legend says that this panel shows the sketches of the mechanism) but to me this is not super clear. Also, the arrows in panel A suggests a preferred direction of the movement but I don't think this is the case.

To address this point, we have modified Fig. 1A and panels F and G. We also added a new supplementary movie 1 illustrating the mechanism.

2) Cryo-EM images are beautiful. Only comment: is there a way to have more “quantitative” information from these images? For example, when the authors say “We analyzed the data set first using 3D classification, which revealed three dominant, structurally distinct 3D classes in the data set (Fig. 3D).” what does this really means? How “dominant” are these three classes? Can the authors define a quantitative value to understand how many of the rotors are in the 3 conformations? Can the authors define what is the statistical significance of these values? This is more a curiosity from my side as I am quite ignorant about Cryo-EM results analysis...

In Figures S15 we give the number of particles in each class. After 2D classification, 165k particles were present. Several runs of 3D classification were performed. Some classes showed fully assembled particles with the rotors in different conformations, while in other classes either the T-cross bar of the camshaft was not visible, or the classes showed defective particles. The three classes with the three different conformations contain 31k, 27k and 20k particles, respectively.

3) Now, the more interesting part is obviously the one regarding the characterization of the rotor movement. This is done with fluorophores in different positions. The authors also extended the T-crossbar of the camshaft with a 10-helix bundle lever arm resulting in a ~ 290 nm long “pointer”. Authors suggest that this “amplifies and also slows down the motions of the camshaft due to friction with the solvent to facilitate tracking the camshaft motions in real-time”. Can the authors elucidate this? What is the effect of this elongation on the rotor movement speed? According to the sentence reported above one would expect that the speed of the rotor could be modulated at will by using longer or shorter arms...can the authors explain this better in the text and report some demonstration of this? For example, do the authors have attempted the same fluorescence experiments without an elongated arm?

The T-cross bar by itself is ~60 nm long (i.e., the radius of rotation is ~30 nm). Since the pixel size of our camera is ~100 nm, we would not be able to clearly detect a rotation with such a small radius, given the frame rate of our camera. The diffusion coefficient can be written as $D = kT/\gamma$, where γ is the friction coefficient. For a rotating cylindrical particle γ is proportional to L^3 , with L length of the cylinder. This means that the diffusion coefficient of a particle of $L = 60$ nm (short rotor) is ca. 150x higher than for a particle with $L = 320$ nm (long rotor).

Ketterer et al. previously measured a simpler rotating mechanism with different lever arm lengths (radius 120 nm vs 550 nm), finding enhanced rotational diffusion for the shorter variant.

4) From what I understand there is no preferred orientation of rotation (Brownian diffusion without directional bias). It would be nice to understand if this is totally expected or if there could be some bias due, for example to the composition of the rotor (DNA sequences) or other factors. Also, could a direction be designed on purpose? This could be interesting.

In the absence of energy supply the bias must vanish in accordance with the laws of thermodynamics. To impart effective directionality on the motion of the rotor one can rely on the framework of Brownian ratchets to turn the rotors into motors. For example, some additional process such as a dissipative chemical reaction or external field could be used to dynamically modulate the energy landscape in which the rotor moves. We consider our mechanical device presented in this manuscript as a crucial milestone toward implementing and testing such modulation in the future. See also our reply to points 7-9.

5) From Figure 4D and 4F it appears that one conformation is preferred over the others. Actually the three seem to have all different statistical distributions. Can the authors explain this? Could it be that the interaction between Cy3 in the arm and Cy5 (in a specific stator) affects the statistical distribution of the rotation? Or this is due to other reasons?

One of the stator unit's pawls is bound to the surface, which probably reduces its motility. Although the motion of the camshaft and the pawl are relative to each other, this bound pawl represents a stiffer obstacle for the camshaft rotation. This might be the cause of the different statistical distribution. Moreover, although the global shapes of the pawls are very similar, there are many local structural differences that could influence the rotor dynamics. For example, the base-stacking sequence between the bearing and the pawls of each monomer is different. This might also influence the mobility of the pawls, hence the rotation of the camshaft. The cy3 and cy5 are far away from each other (at least ca. 200 nm), so we think it is unlikely that the interactions between the dyes influence the dynamics.

6) "Particles can apparently rotate with speeds in the range of up to 3 revolutions per second". Can the authors clarify this sentence? How was this calculated? This also refers back to my comment n. 3.

The angular velocity was calculated as the angular difference between two consecutive frames divided by the time in one frame. We then calculated a histogram of all the angular velocities and plotted it (see Fig. 4H). The maximum angular velocity is around 1500°/s, which actually corresponds to 4 revolutions (we mistakenly wrote 3 revs in the previous manuscript version, apologies for confusion caused).

7) In more general terms the rotor cannot be really defined as a machine: the motion is brownian and thus really the contrary of a machine (maybe I am too drastic in this...). Can the authors comment on this in the conclusions?

We agree that our object does not meet the definition of a machine quite yet, because our rotor does not convert input power in a directional way. Therefore, we have been careful to describe our system as a “mechanism” and “device” and not labeling it as a machine. We examined our manuscript again to make sure the distinction is clear to the reader. However, our mechanism does a crucial step toward building machines because for the first time it allows coupling the motion of movable parts in a designed nanoscale system. We edited the concluding paragraph (see next point and reply to point 9).

8) The authors also say that “it could be of interest to direct the opening/closing transitions in the stator by the consumption of chemical fuel (6-12) to create a chemically fueled rotary nanomotor”. However, this should also be coupled with a preferred direction of the rotor. Maybe this should be discussed by the authors (see previous point).

This question brings us back to the framework of Brownian ratchets. Strategies for imparting directionality on the motion of molecular motors has been discussed in depth for example by Astumian and Stoddart (1, 2). The preferred direction would come from the direction of chemical reaction toward equilibrium. We added these citations to the manuscript to provide more context to the reader. See also our reply to point 9.

**9) Also, and more importantly in my opinion, another aspect should be discussed in the conclusions. Living machines often (almost always) rely on non-equilibrium processes fueled by chemical energy. This makes it possible to perform work and repeated tasks. A continuous supply of chemical energy is required to maintain biological machines in an active state, giving rise to life-like properties such as growth, motion, communication, and adaptation (some of these functions have been described in the introduction). The utilization of energy-dissipating mechanisms, could bring the field of DNA nanotechnology and of DNA machines closer to building “living” devices and materials. It would be good to discuss this in the conclusions. While different dissipative DNA switches and DNA structures have been described recently, dissipative DNA origami machines have yet to be described (I think)!
Francesco Ricci**

Indeed, our system moved strictly in thermal and chemical equilibrium. Moving these mechanisms away from equilibrium will be a rich and interesting ground of research for the years to come. Given that reviewer 4 requested shortening the manuscript, we have decided against further in-depth discussion of work that lays ahead. Our concluding paragraph now reads as follows:

Our mechanism operates through coordinated motion of its components. As such it could provide a framework for creating artificial nanomachines based on the concept of Brownian ratchets (33, 34). For example, the opening/closing transitions of the pawls in the stator could potentially be gated by the consumption of chemical fuel (6-12) to create a chemically fueled rotary nanomotor. Likewise, due to microscopic reversibility, it is conceivable that such a system could potentially be reversed and used for uphill chemical synthesis, as in ATP synthase, by applying mechanical torque to the central shaft, thus creating a chemical generator. In that pursuit, the coordinated motions of the pawl and the camshaft, or new variants of it, could be employed to cyclically bring reactants into close proximity. All of these applications require the creation of intricately shaped components and their assembly into a functional mechanism. Our work shows a route for how such tasks can be accomplished but also highlights the challenges involved in imparting the desired functionality on such ultraminiaturized molecular mechanisms. We expect that the realization of more complex artificial machinery will go hand in hand

with further improvements in analyzing continuous molecular motions by cryo-EM (30, 35) and with improved predictive computational approaches (32, 36).

Reviewer #3 (Remarks to the Author):

Dear Editor,

In this impressive work titled “A nanoscale reciprocating rotary mechanism with coordinated mobility control”, inspired by biological ATP-synthase rotary motor, Bertosin et al. developed a passive rotary device made of DNA origami, demonstrating, to my opinion, the most intricate rationally designed DNA origami dynamical structure fabricated thus far, and significantly advanced several key tools and features necessary for the development of DNA nanoengineering, a field that has already radically transform our ability to fabricate and control bioinspired nanomachines and devices. Specifically, using clever blunt ends interactions, a method invented previously by the group, the authors combined four different origami constructs (six in some cases) to create a passive rotary device that demonstrate long distance allosteric coordination (dozens of nanometers) between a rotating shaft and three stationary pawls that help holding the shaft in place, resembling the allosteric coordination observed in ATP-synthase. Allosteric coordination enables mechanical transfer of information across different sections of a given molecular complex, and it is therefore essential for the proper function of many biological machines and enzymes in which different sections of the molecular complex that perform different tasks needs to operate in coordination. The authors were able to demonstrate that rational design of DNA origami structures according to engineering rules and experimental and computational practices developed by the DNA nanotechnology community in recent years, many of which by the authors research groups, can reproduce such intricate large-distance coordination in DNA based synthetic complexes.

The long-distance allosteric control was demonstrated by showing that changes in the type and strength of a set of orthogonal interactions between pawls and between them and the bearing element (but not with the shaft), can controlled the shaft rotations speed, thus, information about the dynamics of one part of the complex influence the behavior of another part in a rational, controlled and understood way and (generally) as designed, to my opinion, a major achievement.

Furthermore, the authors smart usage of Cryo-EM and single-molecule fluorescence to monitor the fabrication of the rotary complex and to resolve and understand its complex structural and dynamical behavior (together with the usage of MrDNA computer simulations), is an excellent demonstration of how to further advance the field.

The work contains several design and experimental elements that are worth mentioning. The design complex is probably the most tight-fit origami structure demonstrated thus far, and it contains more intricate different structural elements (such as a rotating shaft and its bulging helical element that prevents shaft dissociation, three different moving pawls, a hollow bearing, several blunt-ended helical elements that lock the origami complex together, a 290 nm pointer) than any other DNA complex I am familiar with. Successfully putting all these elements together is impressive. Further, the excellent usage of Cryo-EM and single-molecule fluorescence imaging to resolve the rotary motor complex structure and dynamics. With these tools the authors monitored the rotor fabrication process and characterize the rotor dynamics including shaft states and the shaft angle and rotation rates distributions and demonstrated the expected Brownian dynamics. Taken together, I consider the above list a major achievement that significantly promote the field on several fronts.

Finally, I completely agree with the authors suggestion that the passive rotary device and the allosteric function presented here are major steps towards effective

nonautonomous, and finally, autonomous chemical-energy driven rotary motors, a goal considered a 'holy grail' in the field of artificial molecular machines.

The analysis of the results seems correct, and the manuscript and the figures are generally clear (see the 'minor comments' section below).

For the reasons detailed above I consider this an excellent work and I am sure that it will be interesting for both experts and more general audience alike and that it should be published in Nature Communications.

Below are two minor comments that may help improve the manuscript.

Minor comments:

1. This is a minor comment, but I am not sure that the velocity distribution should behave in a gaussian fashion (the angle distribution should indeed be gaussian), rather I think it should behave as two exponents (one for the negative and the other for the positive velocities), and the seemingly gaussian distribution observed is a result of the method inherent limitation to resolve the high frequency of low-velocities events. In other words, the experimental/instrumental response function (IRF) smooths the exponents to look somewhat like a gaussian in low velocities, and the long tails towards high positive and negative velocities are explained by the two exponents long tails. If this is correct the experimental results fit even better to the theory, demonstrating even higher data quality. Please check.

To address this comment, we fitted the data with a gaussian and with an exponential for the positive and negative velocities.

A
RMSE: 0.018

B
RMSE: 0.025

C
RMSE: 0.026

The gaussian describe the trends of the data better than the exponential fits in terms of the total RMSE. In an ideal gas, the velocity distribution of individual atoms along a particular cartesian direction is expected to follow a gaussian (Maxwell-Boltzmann distribution); by analogy, the rotor angular velocity distribution is expected to be a gaussian if the rotor moved within a practically flat energy landscape. In another unrelated work we did in fact observe a non-gaussian velocity distribution in a translational diffusive mechanism which could be traced back to the unusual shape of the underlying energy landscape (3).

2. In some places the text is not immediately clear and is too long (mainly in the results section). Some more editing may improve the manuscript. Excellent work.

We revised the manuscript to improve clarity as suggested and the manuscript was shortened by 346 words.

Reviewer #4 (Remarks to the Author):

The manuscript describes the design and fabrication of DNA structures that can assemble to form a rotary complex. The authors point to inspiration from rotary ATPases such as the ATP synthase as biological examples of remarkable molecular motors. Similar to ATP synthase, the authors design a three part stator and a rotor that can rotate inside the stator complex. However, unlike the ATP synthase, the synthetic complex does not couple rotary motion to chemical or mechanical work. Despite so, the work provides an important step towards being able to make nanoscale molecular motors and the authors characterized their designs using a variety of biophysical tools.

1. The authors do not provide quantification of the amount of structural deformation observed during rotation of the rotor. How much are the pawls displaced as the rotor rotates? Is this deformation different from the displacement observed during simple Brownian motion of the pawls? This can probably be estimated from the cryo-EM structures, and will provide an indication of how well chemical and mechanical work can be coupled to rotary motion as the authors indicate in the discussion.

We used MultiBody analysis of the whole complex with released camshaft to address this point. We treated each pawl as a separated rigid body and performed principal component analysis with RELION. In two modes we found movements of the pawls, represented in the figure below. For clarity, we displayed only the extremes of the movements. The movements of the pawls range from ~2 nm to ~28 nm.

Figure S18 Principal component analysis of the pawls. (A) The two 3D structures (gray and blue) represent the extreme movements of the pawls found in one of the eigenmodes. (B) Same as (A) but for another eigenmode.

We added the figure in the supplementary information file.

2. The entire rotary complex is very flexible, as indicated by the cryo-EM data and depicted in the videos. In the most flexible variants, the pawls seem almost randomly displaced. The authors show that flexibility of the pawls can be tuned to increase stiffness, but that also seems to prevent mechanical displacement of the pawls. Is there a middle ground that allows the pawls to remain stationary in the ground state, but be flexible enough to be displaced during rotary action? Additional plots/figures showing stiffness to pawl displacement might help to tease this out.

Our coarse-grained simulations indicate that, when forced to rotate, the pawls of fully closed variant (v6) are displaced only slightly by about ~2.5 nm to allow the cam to pass by, suggesting that pawls must easily be displaced by this distance to allow rotation. A more flexible rotor (v1) exhibited pawl displacements of only ~5 nm on average. In fact, we have

already designed a rotor that minimally displaces the pawls during rotation producing a shift of only ~2.5 nm, see Fig S33C, bottom row. The increased stiffness of the rotor does slightly reduce its rotation rate as can be seen from the experimental RMSD. It remains possible that additional design modifications could result in a rotor with similarly immobile pawls but even faster rotation, but our simulations suggest that variant 3 is already almost as immobile as possible while still allowing rotation. Related to this topic, we have added a new principal components analysis of the simulation trajectories that highlights the dominant motion of the pawls of variants 1, 3 and 6 in the supplementary information as Fig. S34.

3. While the authors do not show in this study how chemical and mechanical work could be precisely coupled to rotary motion, this point should be discussed in more detail in the discussion section.

Please refer to our replies to related questions by reviewer 2.

Minor points:

-Structures in figures need scale bars.

We added scale bars in Fig. 2 and 3.

-Figure 2 – Panel G is misleading, it seems like the structure of the stator+rotor complex; consider removing panels

-Figure 2H is difficult to interpret

We modified Fig. 2G to make it clearer. We removed Fig. 2H and added instead supplementary movie 2 that morph the trajectory from the empty stator to the assembled complex.

-Figure 3 – how is rotation locked vs released? Described in text but not clear how it works

The locked version is not rotating, while the released version is. We added Fig. S25 (see answer to reviewer 1) to clarify this point.

1. Y. Feng *et al.*, Molecular Pumps and Motors. *J Am Chem Soc* **143**, 5569-5591 (2021).
2. R. D. Astumian, Trajectory and Cycle-Based Thermodynamics and Kinetics of Molecular Machines: The Importance of Microscopic Reversibility. *Acc Chem Res* **51**, 2653-2661 (2018).
3. P. Stömmer *et al.*, A synthetic tubular molecular transport system. *Nat Commun* **12**, 4393 (2021).

REVIEWERS' COMMENTS

Reviewer #1 (Remarks to the Author):

The authors addressed the minor points well. However, the major shortcoming (lack of novelty) was neither considered nor discussed in this revised version of the manuscript. The lack of novelty is not "on the grounds of our system moving between three preferred states" but that a conceptually almost identical DNA nanorotor was published by Ahmadi et al. already. However, due to the high degree of detail about the novel nanorotor provided in the current study, I recommend publication in a sister journal.

Reviewer #2 (Remarks to the Author):

No further comments from my side. The authors responded well to all my comments and the manuscript is ready for publication in my opinion.

Reviewer #3 (Remarks to the Author):

Dear Authors,

I agree with the authors answer to my first (minor) comment/suggestion, they are of course right, and I was wrong (funny, because I actually teach exactly that in elementary Phys-Chem course). I believe this work is a significant step in the efforts to understand and fabricate artificial molecular machines with properties mimicking that of biomolecular machines, and agree with the authors respond to the criticism raised by Reviewer 1.

I therefore think the manuscript is ready for publication.

Again, an excellent work.

Reviewer #4 (Remarks to the Author):

1. The authors provide a new supplementary figure (Figure S18) showing displacement of the pawls from MultiBody analysis of their data. What was the input data? The authors need to more clearly state what is being compared, otherwise "a range of 2-28 nm" has no real meaning. The authors indicate in Figure S18 that some of the pawls move ~26-28 nm, but which part moves 2nm? Is the movement from Brownian motion or rotation of the shaft? How well correlated is the movement of the pawls to the movement of the camshaft? More details need to be provided.

2. Figure S33C, which color corresponds to which variant?

3. Figure S34, images and labels are cut off. Need scale bars.

REVIEWERS' COMMENTS

We thank the reviewers again for evaluating our revised work. We have addressed all comments in our point-by-point response below.

The reviewer comments are given in **bold face**, the author responses are in plain text, while revised manuscript text are quoted in times new roman.

Reviewer #1 (Remarks to the Author):

The authors addressed the minor points well. However, the major shortcoming (lack of novelty) was neither considered nor discussed in this revised version of the manuscript. The lack of novelty is not “on the grounds of our system moving between three preferred states” but that a conceptually almost identical DNA nanorotor was published by Ahmadi et al. already.

However, due to the high degree of detail about the novel nanorotor provided in the current study, I recommend publication in a sister journal.

We are puzzled by this comment, we believe we discussed at length the novel concepts that are shown in our manuscript, in addition to of course all the novel, so far unpublished data. We respectfully disagree with the reviewer's statement that a “conceptually almost identical DNA nanorotor” had been published before. To see this, it suffices to have a quick look into the work by Ahmadi. Not only are the structures completely different, also the driving mechanism and the depth of experimental analysis are completely different. Furthermore, we note that the reviewer offers contradictory points here, in one sentence he/she claims our work were not novel, in the last sentence he/she labels our nanorotor as novel.

Nonetheless, we noticed that we have overlooked to actually cite the work by Ahmadi et al in the introduction where we cover previous work in the field. We apologize for this oversight and have now added a reference to Ahmadi et al. work in our manuscript.

Reviewer #2 (Remarks to the Author):

No further comments from my side. The authors responded well to all my comments and the manuscript is ready for publication in my opinion.

We thank Reviewer #2 for the positive evaluation of our manuscript.

Reviewer #3 (Remarks to the Author):

Dear Editor and Authors,

I agree with the authors answer to my first (minor) comment/suggestion, they are of course right, and I was wrong (funny, because I actually teach exactly that in elementary Phys-Chem course).

I believe this work is a significant step in the efforts to understand and fabricate artificial molecular machines with properties mimicking that of biomolecular machines, and agree with the authors respond to the criticism raised by Reviewer 1.

I therefore think the manuscript is ready for publication.

Again, an excellent work.

We thank Reviewer #2 for the positive comments on our work.

Reviewer #4 (Remarks to the Author):

1. The authors provide a new supplementary figure (Figure S18) showing displacement of the pawls from MultiBody analysis of their data. What was the input data? The authors need to more clearly state what is being compared, otherwise “a range of 2-28 nm” has no real meaning. The authors indicate in Figure S18 that some of the pawls move ~26-28 nm, but which part moves 2nm? Is the movement from Brownian motion or rotation of the shaft? How well correlated is the movement of the pawls to the movement of the camshaft? More details need to be provided.

The input data for Figure S18 were all the particles in the three 3D classes that showed a different conformation of the camshaft (see maps in Figure S15D). Starting from these maps, 10 rigid bodies were defined: the camshaft, the 6 pawls (2 per each stator unit) and the 3 bearings (1 per each stator unit). With MultiBody and principle component analysis we could identify the main motions in this particle dataset. In this way, we could identify two main motions of the pawls (depicted in S18 A and B, respectively). We then picked the two extremes for each motion and overlapped them (Fig. S18, where the two extremes are in gray and blue, respectively). We identified one helix in each pawl and measured the distance to the same helix in the other extreme conformation. The distances were calculated using UCSF Chimera. As shown in Fig. S18, some pawls are subjected to bigger motions (e.g. 26/28 nm) while other pawls are almost overlapping in the two extremes, (i.e. the distance is around 2 nm). We believe that these motions are a mixture between Brownian motion and interaction with the camshaft. Using this method, we could unfortunately not resolve the dependence between the pawls' motion and the position of the central rotor.

We edited the caption of Figure S18 for clarity.

2. Figure S33C, which color corresponds to which variant?

As shown in the legend in Figure S33C, blue corresponds to v1, orange to v3 and green to v3. We changed the caption accordingly for clarity:

(C) Comparison of forced rotation of variants 1 (blue), 3 (orange) and 6 (green).

3. Figure S34, images and labels are cut off. Need scale bars.

In our version of the Supplementary Information, the images and labels are not cut off; perhaps there was an error when the materials were processed for referees? In any case, we have added scale bars to Figure S34, and we will carefully examine all proofs to ensure that all figures appear intact.